# Improving the TROPOMI CO data product: update of the spectroscopic database and destriping of single orbits

Tobias Borsdorff[1], Joost aan de Brugh[1], Andreas Schneider[1], Alba Lorente[1], Manfred Birk[2], Georg Wagner[2], Rigel Kivi[3], Frank Hase[4], Dietrich G. Feist[5,6,7], Ralf Sussmann[8], Markus Rettinger[8], Debra Wunch[9], Thorsten Warneke[10], and Jochen Landgraf[1]

[1]Netherlands Institute for Space Research, SRON, Utrecht, the Netherlands
[2]Remote Sensing Technology Institute, DLR, Oberpfaffenhofen, Germany
[3]Finnish Meteorological Institute, FMI, Sodankylä, Finland
[4]Institute of Meteorology and Climate Research (IMK-ASF), Karlsruhe Institute of Technology, Karlsruhe, Germany
[5]Ludwig-Maximilians-Universität München, Lehrstuhl für Physik der Atmosphäre, Munich, Germany
[6]Deutsches Zentrum für Luft- und Raumfahrt, Institut für Physik der Atmosphäre, Oberpfaffenhofen, Germany
[7]Max Planck Institute for Biogeochemistry, Jena, Germany
[8]Karlsruhe Institute of Technology (KIT), IMK-IFU, Garmisch-Partenkirchen, Germany
[9]Department of Physics, University of Toronto, 60 St. George Street, Toronto, ON M5S1A7, Canada
[10]Institute of Environmental Physics, University of Bremen, Bremen, Germany

**Correspondence:** T. Borsdorff (t.borsdorff@sron.nl)

**Abstract.** On 13 October 2017, the Tropospheric Monitoring Instrument (TROPOMI) was launched on the Copernicus Sentinel-5 Precursor satellite in a sun-synchronous orbit. One of the mission's operational data products is the total column concentration of carbon monoxide (CO), which was released to the public in July 2018. The current TROPOMI CO processing uses the HITRAN 2008 spectroscopic data with an updated water vapor spectroscopy and produces a CO data product compliant with the mission requirement of 10% precision and 15% accuracy for single soundings. Comparison with ground-based CO observations of the Total Carbon Column Observing Network (TCCON) show systematic differences of about 6.2 ppb and single orbit observations are superimposed by a significant striping pattern along the flight path exceeding 5 ppb. In this study, we discuss possible improvements of the CO data product. We found that the molecular spectroscopic data used in the retrieval plays a key role for the data quality where the use of the Scientific Exploitation of Operational Missions - Improved Atmospheric Spectroscopy Databases (SEOM-IAS) and the HITRAN 2012 and 2016 releases reduce the bias between TROPOMI and TCCON due to improved $CH_4$ spectroscopy. SEOM-IAS achieves the best spectral fit quality (root-mean-squared (rms) differences between the simulated and measured spectrum) of 1.5e-10 $\mathrm{mol\,s^{-1}\,m^{-2}\,nm^{-1}\,sr^{-1}}$ and reduces the bias between TROPOMI and TCCON to 3.4 ppb while HITRAN 2012 and HITRAN 2016 decrease the bias even further below 1 ppb. HITRAN 2012 shows the worst fit quality (rms=2.5e-10 $\mathrm{mol\,s^{-1}\,m^{-2}\,nm^{-1}\,sr^{-1}}$) of the tested cross-sections and furthermore introduces an artificial bias of about -1.5e17 molec/cm$^2$ between TROPOMI CO and the CAMS-IFS model in the tropics caused by the $H_2O$ spectroscopic data. Moreover, analyzing one year of TROPOMI CO observations, we identified increased striping patterns by about 16 % percent from November 2017 to November 2018. For that, we defined a measure $\gamma$ quantifying the relative pixel-to-pixel variation of CO in cross and along track direction. To mitigate this effect, we discuss two destriping methods applied to the CO data a posteriori. A destriping mask calculated per orbit by median filtering of the data in the cross-track direction

significantly reduced the stripe pattern from $\gamma = 2.1$ to $\gamma = 1.6$. However, the destriping can be further improved achieving $\gamma = 1.2$ deploying a Fourier analysis and filtering of the data, which corrects not only for stripe patterns in cross-track direction but also accounts for the variability of stripes along the flight path.

## 1 Introduction

The Tropospheric Monitoring Instrument (TROPOMI) is the single payload of the Copernicus Sentinel-5P satellite that was launched by the European Space Agency (ESA) on 13 October 2017. The instrument provides spectral measurements of the solar radiance reflected by Earth and its atmosphere in the ultraviolet-visible (UV-VIS, 270-495 nm), near-infrared (NIR, 675-775 nm), and the shortwave-infrared (SWIR, 2305-2385 nm) (Veefkind et al., 2012). The novelty of the mission is the daily global coverage, the high spatial resolution of 3.5x7 $km^2$ or 7x7 $km^2$ depending on spectral range, and the higher signal-to-noise ratio (SNR).

One of the primary goals of the mission is to measure the total column concentration of carbon monoxide (CO) in Earth's atmosphere. CO is a trace gas emitted by incomplete combustion (e.g. biomass burning, traffic, and industrial activity) and its only sink is the reaction with the hydroxyl radical (OH) (Spivakovsky et al., 2000). Due to its relative low background concentration and its moderate lifetime (Holloway et al., 2000), it is established as a tracer for anthropogenic air pollution and the atmospheric transport of pollutants on local, regional and global scales.

The TROPOMI CO data product is retrieved from the SWIR measurements of the TROPOMI instrument (Landgraf et al., 2016a, b). Early in the mission, Borsdorff et al. (2018b) inter-compared the TROPOMI CO column with the simulated CO fields of the Copernicus Atmosphere Monitoring Service - Integrated Forecasting System (CAMS-IFS) released by the European Centre for Medium-Range Weather Forecasts (ECMWF). Furthermore, Borsdorff et al. (2018a) validated the product with ground-based Fourier Transform (FTS) measurements from selected sites in the TCCON network which resulted in the release of the TROPOMI CO data product by ESA in July 2018. The analysis of Borsdorff et al. (2018a) showed a significant difference between the TROPOMI CO data product with the ground-based validation measurements of the TCCON network of about 6.4 ppb. Here, the bias between TROPOMI and the TCCON CO measurements was used to estimate the product accuracy and the scatter in the difference between both measurements indicated an upper boundary for the precision of the TROPOMI instrument. This study also showed that stripe patterns along the flight path in the TROPOMI CO data for single orbits can exceed 5 ppb (Borsdorff et al., 2018a) which could hamper e.g. the detection of pollution hotspots and emission estimates. Moreover, the comparison of the TROPOMI and the CAMS-IFS CO datasets indicated a latitudinal difference which represents a problem for the assimilation of the product (Borsdorff et al., 2018b; Inness et al., 2019).

In this study, we discuss in detail the open issues of the TROPOMI CO data product and possible mitigation strategies. Section 2 introduces the TROPOMI CO data, the CO validation measurements of the TCCON network and the CO CAMS-IFS data. In Sect. 3.1, we discuss the use of different molecular spectroscopic databases, the induced biases between TROPOMI CO and the TCCON measurements and the latitudinal dependent bias between TROPOMI CO and the CAMS-IFS model.

Section 3.2 discusses two methods for the stripe correction of single TROPOMI CO orbits. Finally, Sect. 4 provides a summary and recommendations for future TROPOMI CO retrieval approaches.

## 2 Data sets

The operational TROPOMI CO data processing deploys the Shortwave-Infrared CO retrieval (SICOR) algorithm that includes
atmospheric light scattering by clouds to retrieve the vertical trace gas columns of CO, $H_2O$, HDO, and $CH_4$ together with
effective parameters describing the cloud contamination of the measurements (cloud altitude $z$ and cloud optical thickness $\tau$).
The theoretical details for the algorithm are described by Vidot et al. (2012); Landgraf et al. (2016a, b). For this study, we
analyze one year of TROPOMI SWIR measurements from November 2017 to November 2018 using the operational SICOR
as used by Borsdorff et al. (2018b, a, 2019).

The radiative transfer and so the data interpretation depends on spectroscopic data to simulate the absorption lines of at-
mospheric trace gases. The operational TROPOMI CO processor uses the line lists of HITRAN 2008 (Rothman et al., 2009)
for the trace gases CO and $CH_4$ and the updated water vapor spectroscopy for HDO and $H_2O$ by Scheepmaker et al. (2012),
who updated the line intensities, pressure shifts and pressure broadening parameters by fitting laboratory spectra of water
vapor (HITRAN 2008+H2O in Table 1). They showed that the $H_2O$ column retrieval from ground-based FTS measurements
is improved by the updated line parameters. Also the HITRAN 2012 release (Rothman et al., 2013) addressed deficiencies
identified in the HITRAN 2008 water vapor line list. Recently, the Scientific Exploitation of Operational Missions - Improved
Atmospheric Spectroscopy Databases (SEOM-IAS) which is an ESA Project revised the line list parameters/absorption cross
sections of $O_3$, CO, $CH_4$, $H_2O$, HDO, and $SO_2$ with the objective to improve the quality of the Sentinel-5P data products
(https://www.wdc.dlr.de/seom-ias/). The $CH_4$ and $H_2O$ line lists of SEOM-IAS were tested by fitting atmospheric spectra
recorded by FTIR spectrometry, resulting in significantly improved residuals in spectral sections dominated by $CH_4$ and $H_2O$
compared to HITRAN 2012 (Hase et al., 2018). Some of the updates from SEOM-IAS regarding the spectroscopy of water
vapor are already integrated in the new HITRAN 2016 release (Gordon et al., 2017).

To test the effect of the different spectroscopic databases on the TROPOMI CO retrieval, we performed multiple retrievals
where we substituted the spectroscopic data used for the operational TROPOMI CO retrieval which is based on HITRAN 2008
with $H_2O$ updated by Scheepmaker et al. (2012), by the one of SEOM-IAS, HITRAN 2012, or HITRAN 2016. Here we
substituted the spectroscopic data for all retrieval species at once but also for each trace gas individually. The remaining
retrieval settings are identical with the ones of the operational processing.

For the different spectroscopies, we validated the TROPOMI CO column densities with the TCCON CO product at several
sites of the network. The TCCON CO columns have an accuracy better than 4 % (Wunch et al., 2015). The geolocation, altitude,
and citation information of the TCCON stations is summarized in Table 2. The validation approach is described in detail by
Borsdorff et al. (2018a). First, we select TROPOMI CO data in a radius of 50 km around a TCCON site and subsequently
corrected for the altitude difference between the TROPOMI ground pixel and the site. Finally, we compare daily averaged
TROPOMI and TCCON CO data of the same day and estimate the scatter in the TROPOMI data. For the validation of the

TROPOMI data, we discriminated clear-sky observations and those with low clouds as described by Borsdorff et al. (2018a). Figure 1 and Fig. 2 give an example of a time series of daily mean dry air CO column mixing ratios XCO deploying the HITRAN 2016 spectroscopic data. The blue and pink symbols indicates collocated data pairs. These are used for further data analysis in this study, whereas all grey data point are discarded. Moreover, to evaluate the quality of the spectral fit for each

retrieval, we consider the root-mean-square difference $\sqrt{\frac{1}{L}\sum_{l=1}^{L}(y_{\mathrm{meas},l} - y_{\mathrm{sim},l})^2}$, where index $l$ indicates the $L$ spectral components of the measurement $y_{\mathrm{meas}}$ and its simulation $y_{\mathrm{sim}}$ after convergence of the retrieval. Finally, for a collocated data pair, we determine the corresponding averaged root-mean-square difference.

For further analysis, we define a set of diagnostic quantities. For each station of our data set, we define the median bias $b_j$ as the median of the difference $\mathrm{XCO}_{ij}^{\mathrm{TROPOMI}} - \mathrm{XCO}_{ij}^{\mathrm{TCCON}}$ between TROPOMI and TCCON XCO daily mean measurements,

where index $j$ identifies the station, and $i$ indicates the pair of collocated daily mean values. Also the corespondent median route mean square difference $rms_j$ is determined. To characterize the scatter in the difference between TROPOMI and TCCON data, we consider the percentile difference

$$\delta P_j = |\frac{P_j(84.1) - P_j(15.9)}{2}| \tag{1}$$

of the bias distribution, which corresponds to the standard deviation of normal distributed parameters but it is more robust

against outliers. Hence, the choice of 84.1 and 15.9 percentiles would be the $\pm 1$ 1-sigma around the mean for a normal curve. Moreover, the global mean bias $\bar{b}$ is the mean bias of all station biases,

$$\bar{b} = \frac{1}{n}\sum_{j=1}^{n} b_j \tag{2}$$

with $n$ the number of stations and the station-to-station bias variation is defined as the standard derivation

$$\bar{\sigma} = \sqrt{\frac{1}{n}\sum_{j=1}^{n}(b_i - \bar{b})} \ . \tag{3}$$

Fig. 3 shows the statistics of the corresponding biases between TROPOMI and the TCCON measurements.

The inter-comparison of the TROPOMI CO retrievals with the CO data of the CAMS-IFS model follows the approach as described in Borsdorff et al. (2018b), where we interpolated the vertical profiles of the model spatially and temporally to the time and geolocation of the ground pixels of TROPOMI. Then we calculated the total column concentration of CO from the model profiles by multiplying them with corresponding total column averaging kernels of TROPOMI that are provided for

each measurement. By that the comparison is free of the null-space or smoothing error contribution (Rodgers, 2000).

## 3    Results

### 3.1    Spectroscopic Databases

The bias between TROPOMI CO and the ground-based validation measurements of the TCCON network depends significantly on the spectroscopic data base used in the retrieval. Using HITRAN 2016 (see Fig. 3) instead of HITRAN 2008 with $H_2O$

updated by Scheepmaker et al. (2012) (see Fig 4), the difference between TCCON and TROPOMI CO is reduced from 6.2 ppb to 0 ppb for clear sky observations and the station-to-station variability of the bias decreases from 2.6 ppb to 1.8 ppb. Also the scatter $\delta P$ of the bias is reduced from 3.6 ppb to 2.6 ppb. Retrievals from cloudy and clear sky observations agree well and show similar improvements, whereas the fit quality represented by the root-mean-squared (rms) differences between the simulated spectrum and the measurement is only slightly improved. Overall, we conclude an improved agreement between the TROPOMI and TCCON observations using the most recent HITRAN data release from 2016.

Table 1 provides the TROPOMI-TCCON mean bias, the scatter, and the rms of the spectral fit residuals when using the current TROPOMI spectroscopic database, the SEOM-IAS, HITRAN 2012 or HITRAN 2016 data base. We found that any of the new spectroscopic databases improves the bias and $\delta P$ of the biases between TCCON and TROPOMI. For SEOM-IAS, the TROPOMI CO retrievals differ by 3.4 ppb compared to the TCCON results. Furthermore, the table also shows the diagnostics when changing the spectroscopy of only one trace gas and keeping the current TROPOMI spectroscopic database for the other species. It clearly indicates that updating the $CH_4$ cross sections is the main reason for the improved CO product. The quality of the spectral fit is only enhanced using the SEOM-IAS spectroscopy (rms=1.5e-10 $\mathrm{mol\,s^{-1}\,m^{-2}\,nm^{-1}\,sr^{-1}}$), HITRAN 2016 provides the same fit quality as our baseline spectroscopy (rms=1.8e-10 $\mathrm{mol\,s^{-1}\,m^{-2}\,nm^{-1}\,sr^{-1}}$) while HITRAN 2012 worsens it (rms=2.5e-10 $\mathrm{mol\,s^{-1}\,m^{-2}\,nm^{-1}\,sr^{-1}}$).

One of the main applications of the TROPOMI CO data is its use in the CAMS-IFS assimilation system to improve chemical weather forecasting. Therefore, non-physical differences between TROPOMI CO product and the CAMS-IFS model must be avoided. To evaluate this, we first aim to mimic the TROPOMI CO validation in Fig. 3 but using CAMS-IFS CO data instead of TROPOMI observations. Therefore, we spatio-temporally interpolated the model profiles to the corresponding TROPOMI clear-sky and cloudy measurements and applied the averaging kernels. Figure 5 shows a mean difference between CAMS-IFS and TCCON of 2.7 ppb for clear-sky condition with a station-to-station variability of 2.7 ppb and a scatter of the bias of 4.9 ppb. We obtain very similar results when using the averaging kernels for cloudy conditions. Therefore, we can conclude that CAMS-IFS agrees well with TROPOMI CO, and with the retrievals from the TCCON network.

Inness et al. (2019) reported a latitudinally dependent difference between TROPOMI CO and CAMS-IFS model. From 28 January to 3 May 2018, TROPOMI CO is biased high compared to CAMS-IFS by $(0.17\pm0.27)\times10^{18}$ molec. $\mathrm{cm}^{-2}$ in the high northern hemisphere, $(0.07\pm0.19)\times10^{18}$ molec. $\mathrm{cm}^{-2}$ in the Tropics and $(0.009\pm0.12)\times10^{18}$ molec $\mathrm{cm}^{-2}$ in the low southern hemisphere. The CAMS-IFS model is known to underestimate CO in the northern hemispheric extra-tropics, particularly in winter and spring time. Hence, part of the bias between CAMS-IFS and TROPOMI can be due to the model but a systematic error in the TROPOMI CO data cannot be excluded. Figure 6 shows the longitudinal averaged difference between TROPOMI and CAMS-IFS CO fields using the current TROPOMI spectroscopic database, the SEOM-IAS, the HITRAN 2012 and 2016 spectroscopy (color coded) for 10 October 2018. Again, we spatio-temporally interpolated the CAMS-IFS CO profiles to the TROPOMI data and applied the TROPOMI averaging kernels to calculate the CAMS-IFS total CO column concentrations. The upper panel of the figure indicates that the differences are largest for the current baseline spectroscopy and HITRAN 2012 while for the SEOM-IAS spectroscopy CAMS-IFS agree best with TROPOMI CO. The relative latitudinal dependence of the differences are shown in the lower panel of Fig 6, which indicates that HITRAN 2016 spectroscopy leads to the smallest

latitudinal dependence of the differences while HITRAN 2012 results in unrealistic deviations between model and TROPOMI observations of about -1.5e17 molec/cm$^2$ due to the involved H$_2$O spectroscopic data of HITRAN 2012.

To conclude, the choice of a spectroscopic database used for the TROPOMI CO retrieval is crucial. When relying on the TCCON measurements as a validation source, the HITRAN 2016 spectroscopy database is the best choice for the TROPOMI

CO retrieval with no significant overall bias to the validation network and the smallest latitudinally dependent difference with the CAMS-IFS model. Overall, the SEOM-IAS spectroscopy improves the TROPOMI CO retrieval similarly as HITRAN 2016 but comes with a small bias compared to the measurements of the TCCON network. It is the only spectroscopy database that improves the fit quality (rms=1.5e-10 $\mathrm{mol\,s^{-1}\,m^{-2}\,nm^{-1}\,sr^{-1}}$) of the TROPOMI CO retrieval and has practically no bias with the CAMS-IFS model. It is important to note that HITRAN 2016 and SEOM-IAS are not completely independent since some

of the updates from SEOM-IAS are already included in HITRAN 2016. For the operational TROPOMI data processing, the HITRAN 2012 database is out of consideration since it worsens the fit quality (rms=2.5e-10 $\mathrm{mol\,s^{-1}\,m^{-2}\,nm^{-1}\,sr^{-1}}$) quality of the TROPOMI CO retrieval and introduces an artificial bias of about -1.5e17 molec/cm$^2$ with CAMS-IFS caused by issues in the water spectroscopy. We could not see this by comparing with TCCON data because not so many stations are available at the equator.

To finally conclude on the most appropriate spectroscopy database, we must keep in mind also the validity of the validation source. Wunch et al. (2015) estimated the accuracy of the TCCON CO product to be better than 4 % and Borsdorff et al. (2016) noted that TCCON is biased high compared to other validation sources like measurements of the Network for the Detection of Atmospheric Composition Change - Infrared Working Group (NDACC-IRWG) and of the In-service Aircraft for a Global Observing System (MOZAIC-IAGOS). Kiel et al. (2016) found a similar disagreement between NDACC-IRWG and TCCON

measurements. Based on the presented analysis, we favor the HITRAN 2016 and SEOM-IAS spectroscopy for the improved TROPOMI CO processing, although a final judgment requires a better harmonization between the different validation sources, in particular between the ground-based networks TCCON and NDACC-IRWG.

## 3.2 Destriping of single orbits

The TROPOMI CO retrievals from single orbits show a significant striping pattern along the flight path, which is a well-known

feature for observations of push-broom spectrometers (e.g. OMI (Boersma et al., 2011) and MODIS (Rakwatin et al., 2007)). Borsdorff et al. (2018a) already reported that the CO stripes can exceed 5 ppb and can hamper, e.g., the detection of small point sources and the estimate of emissions from fire plumes. The origin of the stripy pattern is not yet understood and is changing with time from orbit to orbit. The TROPOMI level 1 team is optimizing the Calibration Key Data (CKD) to reduce the effect in future. Borsdorff et al. (2018a) suggested an empirical destriping approach that is applied on the CO data fields (see left column

of Fig 7). This method removes first the background of the CO field by a median smoothing in cross-track direction and then determines per orbit a fixed stripe pattern for correction by a median along the flight path. This method already reduces a major part of the stripes in the CO data and is denoted in the following as fixed mask destriping (FMD). Analyzing TROPOMI CO orbit observation, we found that the stripe patterns changes to some extent also along the flight path, which cannot be captured

by this approach. Therefore, we investigate in this study an alternative approach that is based on a Fourier filter destriping (FFD) (see right column of Fig 7).

Transformed domain filtering is widely used in image processing and was already applied for the destriping of MODIS data (Rakwatin et al., 2007). The idea is to transform the TROPOMI CO data $\mathbf{d}$ of one orbit into the Fourier space by the transformation

$$\hat{\mathbf{d}}(\nu_x, \nu_y) = \int\limits_{\infty}^{\infty} \mathbf{d}\, e^{-2\pi i x \nu_x} e^{-2\pi i y \nu_y}\, dxdy. \tag{4}$$

Before this transformation the missing data in $\mathbf{d}$ was replaced by the median value of the corresponding swath and additionally a fixed strip pattern was added to the interpolated missing values deploying the FMD method. Subsequently, the spectral representation of the data $\hat{\mathbf{d}}(\nu_x, \nu_y)$ as a function of the two frequencies $\nu_x$ and $\nu_y$ is multiplied by a filter function $f(\nu_x, \nu_y)$ to remove stripes and then is transformed back by

$$\mathbf{d}_{\mathrm{ds}}(x,y) = \int\limits_{\infty}^{\infty} \hat{\mathbf{d}}(\nu_x, \nu_y)\, f(\nu_x, \nu_y)\, e^{2\pi i x \nu_x} e^{2\pi i y \nu_y}\, d\nu_x d\nu_y. \tag{5}$$

The filter function $f(\nu_x, \nu_y)$ is chosen to filter on high frequencies in cross-track direction (x-dimension) and some low frequencies along the flight path (y-dimension). Hence, this approach removes stripes that have a high frequent part in cross-track and some low frequency change along the flight path. The filter function is defined by

$$f(\nu_x, \nu_y) = 1 - g(\nu_y, 0, \sigma(\nu_x)). \tag{6}$$

Here, $g(\nu_y, 0, \sigma(\nu_x))$ is a collection of Gaussian function for each $\nu_x$ centered around $\nu_y = 0$ with a standard deviation $\sigma(\nu_x)$ which depends linearly on $\nu_x$ as shown in Fig. 8 with $\sigma_{min} = 0.3$ for low frequencies and $\sigma_{max} = 7$ for high frequencies. Here, no filtering was applied for $\nu_x \in [-7, 7]$. These parameters were chosen empirically such that the median of the destriped TROPOMI CO data from one orbit is deviating by less than 1 percent from the original one. Finally, the destriping mask is calculated by $s = \mathbf{d} - \mathbf{d}_{\mathrm{ds}}$.

To measure the effectiveness of the destriping approach, we defined the characteristic

$$\gamma = \frac{std(Dx(\mathbf{d}))}{std(Dy(\mathbf{d}))} \tag{7}$$

where the operator $Dx(\mathbf{d}) = \frac{\partial \mathbf{d}}{\partial x}$ is the discrete derivative operator in cross-track direction (see Fig 9a ) and $Dy = \frac{\partial \mathbf{d}}{\partial y}$ the discrete derivate operator along flight (see Fig 9b) and the function std is the operator to calculate the standard deviation. The derivative $Dy(\mathbf{d})$ represents mostly the natural pixel-to-pixel variability of the measured CO field, whereas $Dx(\mathbf{d})$ is sensitive to the stripe pattern along the flight path. Figure 9c shows $Dx(\mathbf{d}_{\mathrm{ds}})$ when applying the FMD method and Fig 9d when applying the FFD approach. While the FMD method still leaves remaining stripes in the data the FFD approach is more efficient.

For the original data $\mathbf{d}$, $\gamma$ is usually greater than one since the stripes enhance $Dx(\mathbf{d})$ compared to $Dy(\mathbf{d})$. Hence, we expect that the destriping reduces $\gamma$, with $\gamma = 1$ for an isotropic pixel-to-pixel variation in the CO field. However, we cannot demand

$\gamma = 1$ after destriping because different synoptic variation in CO in both directions on average cannot be precluded. A tuning of the destriping algorithm to fulfill $\gamma = 1$ may result in a unwanted smoothing of the CO data.

Figure 10 shows the $\gamma$ value of the TROPOMI measurements from November 2017 to November 2018 without applying any destriping (gray line). Hence, we see a trend in the intensity of the striping pattern that increased by about 16 % in the first year of the mission, which may hint at a possible degradation of the instrument. The FMD approach (pink line) significantly reduces the stripe pattern by about 24 % and removes the trend of the original data. Finally the FFD approach (green line) also removes the trend and further improves $\gamma$ by 20 % compared to the FMD method. Here, it is remarkable that the FFD approach shows also a lower standard deviation of the monthly averages which points to a more consistent destriping with time.

For both destriping methods, we found that the TCCON validation (bias, station-to-station variability of the bias, and scatter of the bias) does not significantly change. For the TCCON validation daily averages in a collocation radius of 50 km were calculated. We found that on this scale, the impact of stripes on single orbit data can be neglected. The advantage of destriping the CO data becomes obvious, when we consider CO emission from fires like in Fig. 7. Here stripes can have a significant impact on the estimated emission and the detection limit of this type of events.

## 4  Conclusions

The TROPOMI instrument is operating successfully since more than one year (13th of October 2017) on ESA's Sentinel-5P satellite, where the SWIR measurements provide the total column concentration of CO with daily global coverage and a high spatial resolution of 7x7 km$^2$. Early in the mission it was concluded that the TROPOMI CO dataset fulfills the mission requirements (accuracy $< 15\,\%$ and precision $< 10\,\%$) and the TROPOMI CO data product was released by ESA in July 2018. Previous studies indicated that the TROPOMI CO product is biased high by about 6.4 ppb compared to the ground-based validation measurements of the TCCON network. Moreover, both a latitudinally dependent difference with the CAMS-IFS model and significant stripe patterns of single TROPOMI CO orbits, exceeding 5 ppb occasionally, were reported.

This study showed that the use of the SEOM-IAS, HITRAN 2012, HITRAN 2016 spectroscopic database significantly affects the CO bias between the TROPOMI and TCCON observations and the CO comparison with the CAMS-IFS model. Currently the operational processing of TROPOMI CO data relies on HITRAN 2008 spectroscopy with updates to the $H_2O$ spectroscopy by Scheepmaker et al. (2012) which results in a bias of 6.2 ppb as derived from one year of observations using TCCON observations as a validation reference. Any of the other investigated molecular spectroscopies improves these diagnostics due to improved $CH_4$ absorption lines in the new databases. Here, SEOM-IAS reduces the bias to 3.4 ppb , HITRAN 2012 to -1.6 ppb, and HITRAN 2016 to 0 ppb. We found similar improvements for the station-to-station variability of the biases. Only the SEOM-IAS dataset improves the spectral fit quality (rms=1.5e-10 $\mathrm{mol\,s^{-1}\,m^{-2}\,nm^{-1}\,sr^{-1}}$) while HITRAN 2012 worsens it (rms=2.5e-10 $\mathrm{mol\,s^{-1}\,m^{-2}\,nm^{-1}\,sr^{-1}}$). A comparison with the CO fields of the CAMS-IFS model indicates that HITRAN 2012 creates an artificial bias of about -1.5e17 molec/cm$^2$ around the equator due to erroneous $H_2O$ spectroscopic data. HITRAN 2016 improves the latitudinal dependency of the bias between CAMS-IFS and TROPOMI CO. To finally conclude on the most appropriate spectroscopy database, we also must keep in mind the validity of the validation

source. Borsdorff et al. (2016) noted that TCCON is biased high compared to other validation sources like measurements of the NDACC-IRWG and MOZAIC-IAGOS. Kiel et al. (2016) found a similar disagreement between NDACC-IRWG and TCCON measurements. Based on the presented analysis, we favor the HITRAN 2016 and SEOM-IAS spectroscopy for the improved TROPOMI CO processing. SEOM-IAS was the only spectroscopic database that improved the fit quality (rms=1.5e-10 $\mathrm{mol\,s^{-1}\,m^{-2}\,nm^{-1}\,sr^{-1}}$) of the TROPOMI CO retrieval. However, a final judgment requires a better harmonization between the different validation sources, in particular between the ground-based networks TCCON and NDACC-IRWG.

Another important shortcoming of the current operational TROPOMI CO product is the CO striping of single orbit data. Analyzing one year of TROPOMI data, we found that the intensity of the striping increased from November 2017 to November 2018 by about 16 %, which degrades the quality of the data. Stripes can occasionally exceed 5 ppb and so hamper the detection of CO hotspots and the CO emission estimations from point sources. We discussed two approaches to destripe the TROPOMI CO level 2 data. Applying a destriping approach, which is constant over an orbit, improved the data significantly. Best results were achieved by a destriping approach filtering in the spectral domain of the orbit data. This approach can account for a variation of stripes along the orbit. Both approaches can cope with the time dependent increase in stripiness but the FFD approach achieves a more homogeneous pixel-to-pixel variability of the destriped CO field with time. For both destriping methods, we found that the TCCON validation (bias, station-to-station variability of the bias, and standard deviation of the bias) does not significantly change. For the TCCON validation monthly averages in a collocation radius of 50 km were calculated. We found that on this scale, stripes on single orbit data can be neglected and so we can conclude that the destriping is not introducing additional overall biases when applied on the data.

## 5 Data availability

The TROPOMI CO data set of this study is available for download at ftp://ftp.sron.nl/open-access-data-2/TROPOMI/tropomi/co/. TCCON data are available from the TCCON Data Archive, hosted by CaltechDATA, California Institute of Technology, CA (US), https://tccondata.org/.

*Author contributions.* Tobias Borsdorff, Joost aan de Brugh, Andreas Schneider, Alba Lorente Delgado, and Jochen Landgraf provided the TROPOMI CO retrieval and data analysis. DLR was providing the SEOM-IAS spectroscopy and the TCCON partners provided the validation datasets. All authors discussed the results and commented on the manuscript.

*Competing interests.* The authors declare no competing interests.

*Disclaimer.* The presented work has been performed in the frame of the Sentinel-5 Precursor Validation Team (S5PVT) or Level 1/Level 2 Product Working Group activities. Results are based on preliminary (not fully calibrated/validated) Sentinel-5 Precursor data that will still change. The results are based on S5P L1B version 1 data. Images/data contain modified Copernicus Sentinel data, processed by SRON

*Acknowledgements.* The presented material contains modified Copernicus data [2017,2018] The TROPOMI data processing was carried out on the Dutch national e-infrastructure with the support of the SURF Cooperative. The work contains modified Copernicus Atmosphere Monitoring Service Information [2017,2018]. Neither the European Commission nor ECMWF is responsible for any use that may be made of the information it contains. TCCON observations from ETL are supported by the CSA, CFI, ORF, NSERC, and ECCC. The TCCON stations Garmisch, Izaña, and Karlsruhe have been supported by the German Bundesministerium für Wirtschaft und Energie (BMWi) via DLR under grants 50EE1711A & D.

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

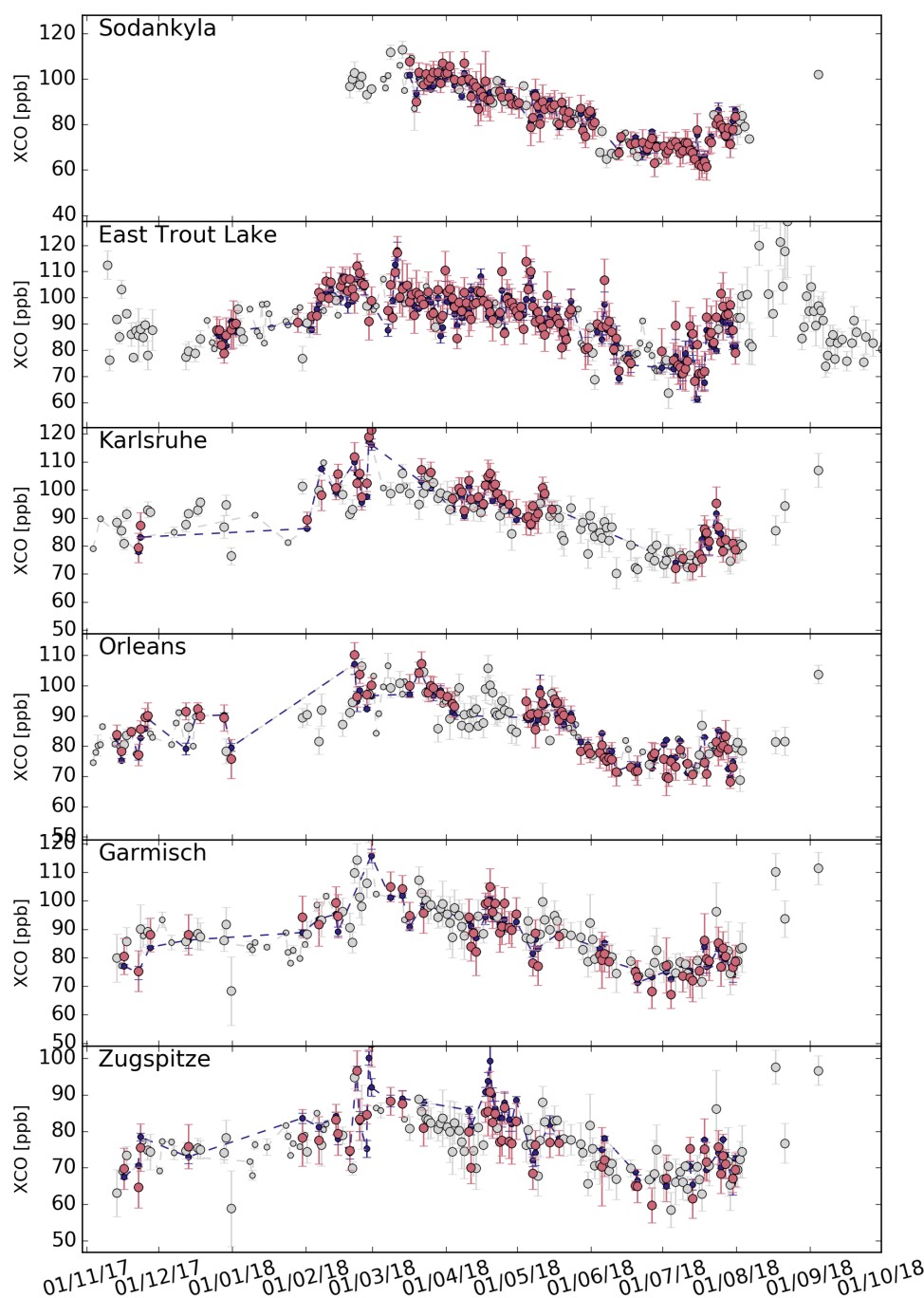

**Figure 1.** Daily means of dry air column mixing ratios (XCO) measured by TROPOMI (pink) and various TCCON stations (blue) under clear-sky and cloudy atmospheric conditions. A co-location radius of 50 km is used. The standard deviation of individual retrievals within a day is shown as an error bar. The retrieval deployed the spectroscopic database HITRAN 2016 for all trace gases. Measurements of both datasets that could not be paired are marked as grey dots (big=TROPOMI, small=TCCON) and are not used in this study.

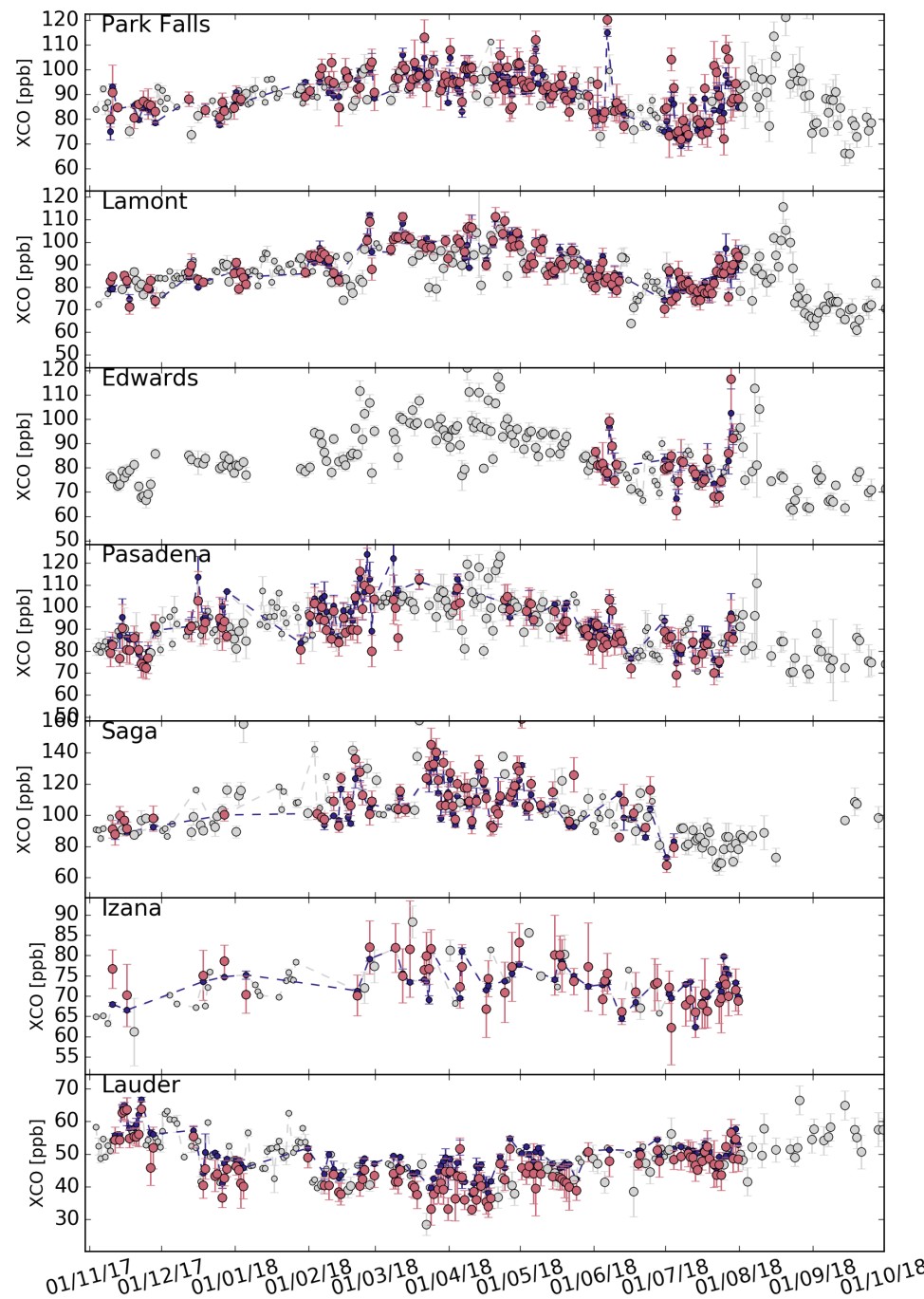

**Figure 2.** As Fig. 1 but for different TCCON stations.

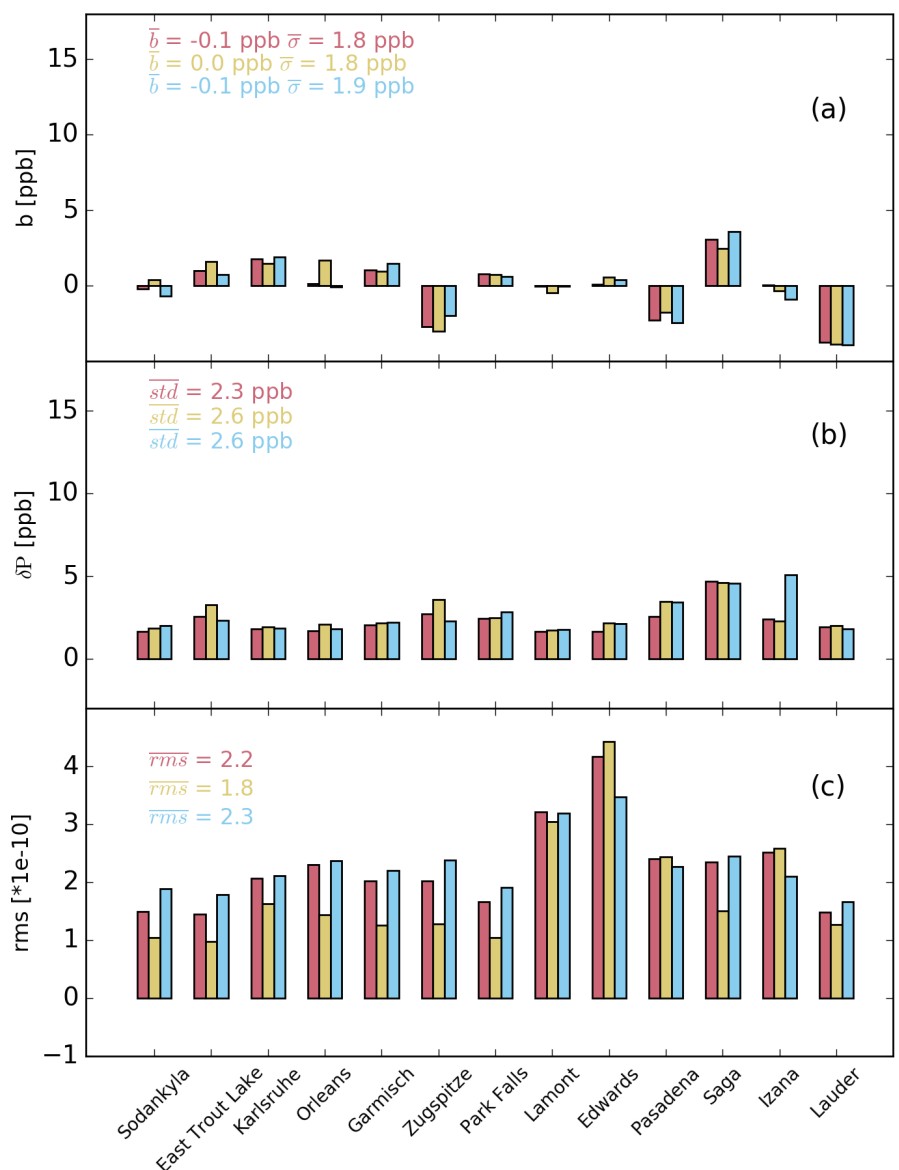

**Figure 3.** (a) median bias $b_j$ (TROPOMI-TCCON) for different TCCON sites between co-located daily mean XCO values of TROPOMI and TCCON (see blue and pink dots in Fig. 1, 2) The global mean bias $\bar{b}$ and the correspsonding standard derivation $\bar{\sigma}$ as defined in Eq. (2) and (3), (b) the scatter $\delta P_j$ of the biases as defined in Eq. (1) with its global mean $\delta \bar{P}$ and (c) the median root-mean-square (rms) of the spectral fit residuals of the individual retrievals per station and its global mean r$\bar{m}$s in $\mathrm{mol\,s^{-1}\,m^{-2}\,nm^{-1}\,sr^{-1}}$. The figure shows TROPOMI retrievals under clear-sky (yellow), cloudy-sky (blue) and the combination of both (pink). No destriping is applied to the TROPOMI data. The retrieval deploys the spectroscopic database HITRAN 2016 for all absorbers.

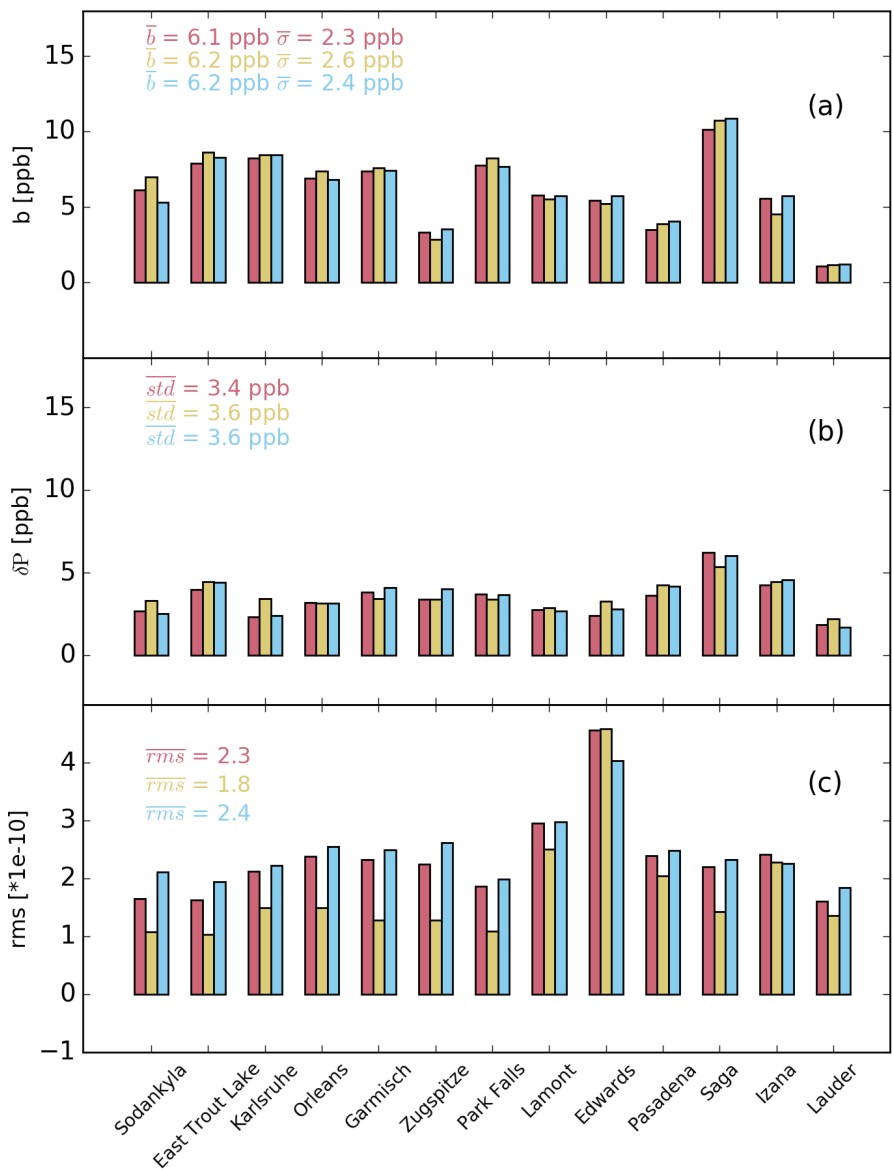

**Figure 4.** Same as Fig 3 but deploying the spectroscopic database used in the operational TROPOMI CO processing (HITRAN 2008 with H$_2$O updates).

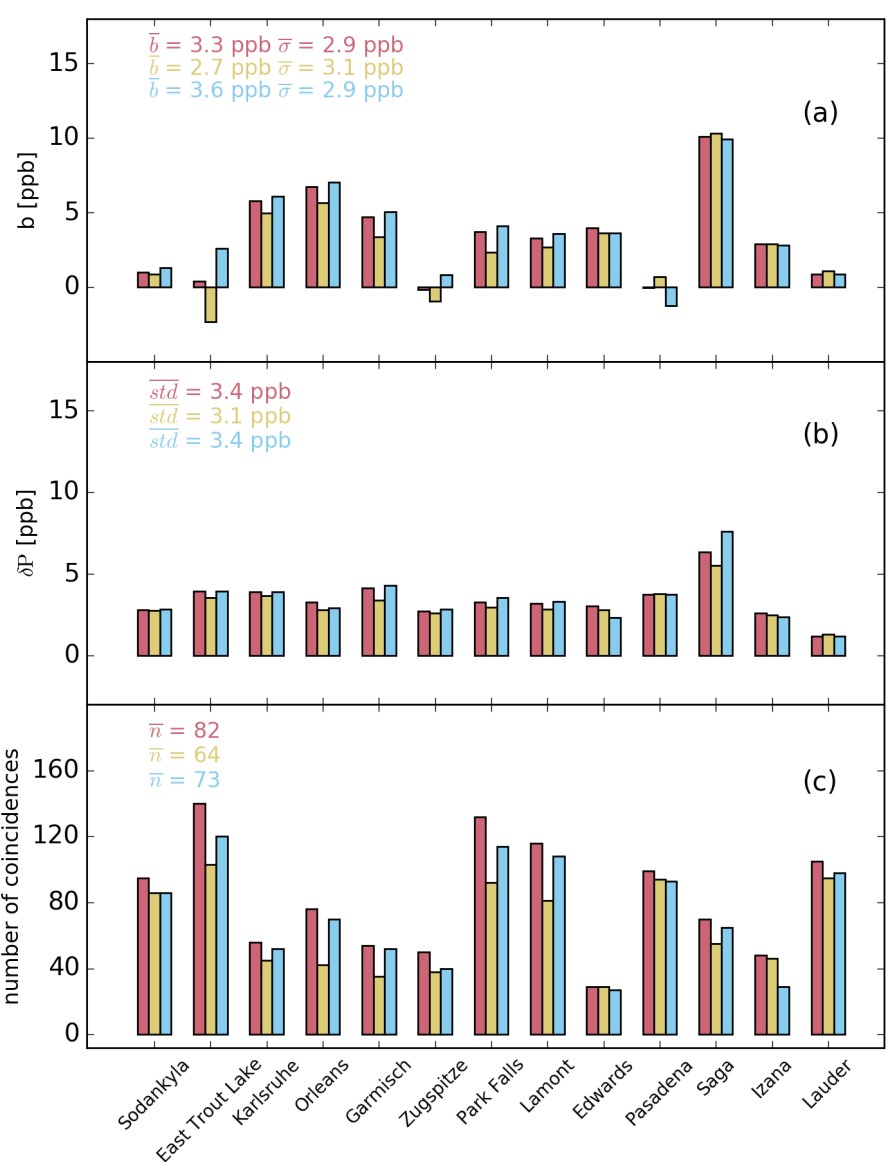

**Figure 5.** Same as Fig 3 but comparing TCCON measurements with CAMS-IFS CO model data, which are co-located with the TROPOMI observations of Fig. 3. To this end, we interpolated the CAMS-IFS model temporally and spatially to TROPOMI measurements and also applied the averaging kernels of TROPOMI on the vertical profiles of the model. In this model comparison a spectral fit quality (rms) plot is not need and therefore replaced by the number of coincidences.

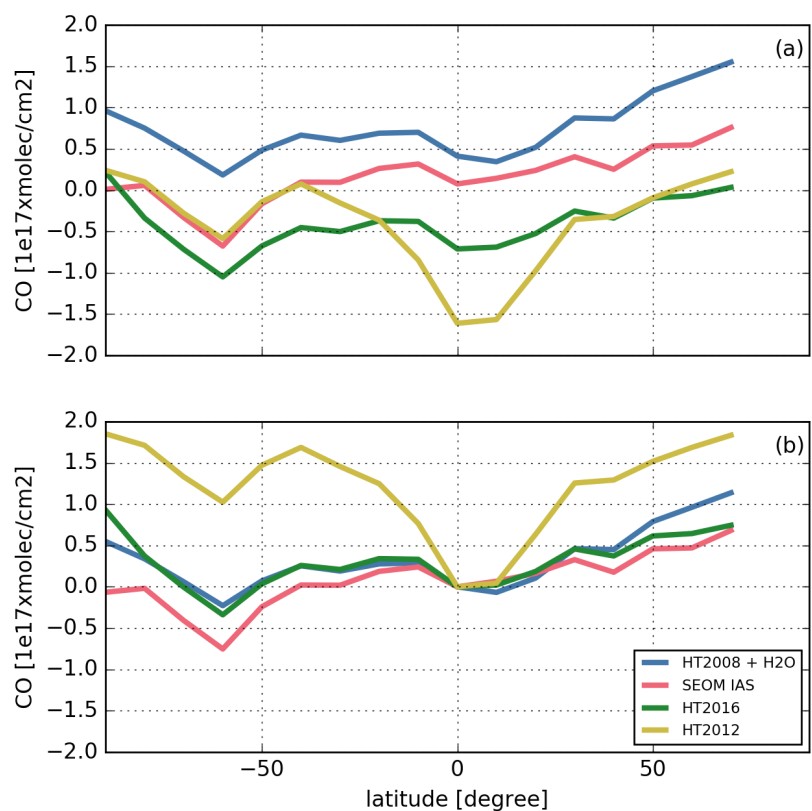

**Figure 6.** (a) Longitudinal averaged difference between TROPOMI and CAMS-IFS model data for 10 October 2018 (TROPOMI-CAMS-IFS). The CAMS-IFS model are spatio-temporally interpolated to the TROPOMI measurements and averaging kernels are applied. The colors indicate the bias when using different spectroscopic databases in the TROPOMI retrieval. (b) Same as (a) but relative to the corresponding difference at $0^o$ latitude to visualize the different gradients in latitude.

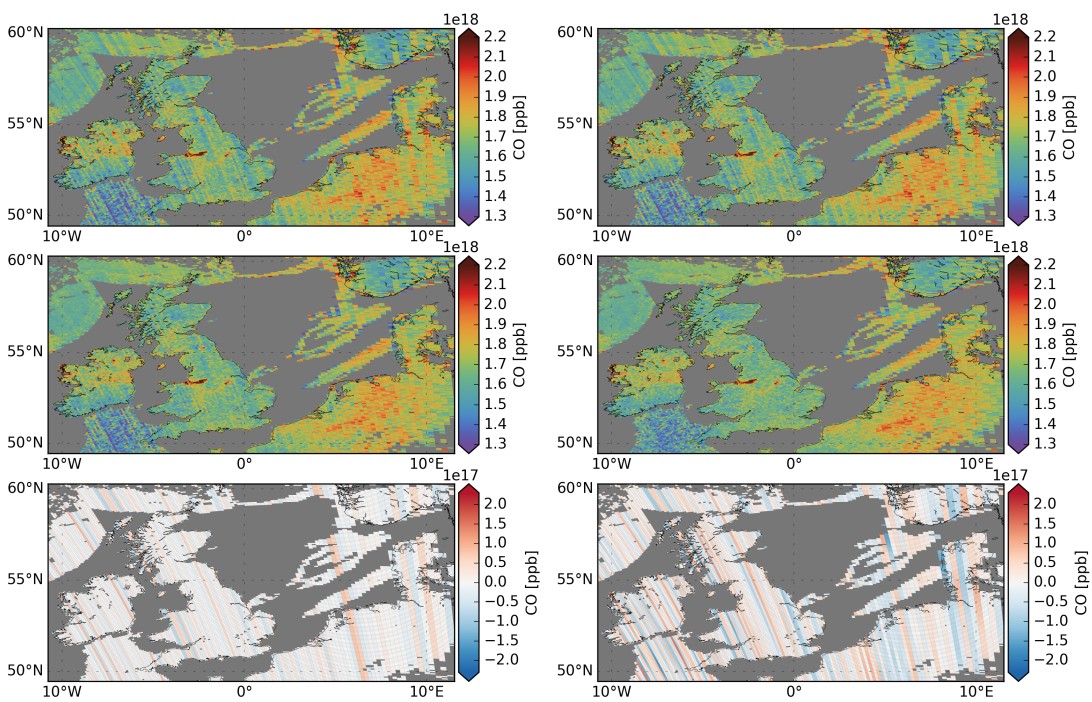

**Figure 7.** CO retrievals of a TROPOMI orbit granule on 27 June 2018 over the UK. Panels of the first row depict the original data, the second row shows the destriped TROPOMI CO data (FMD method left, FFD method right), and the third row illustrates the destriping mask that was subtracted from the original TROPOMI data.

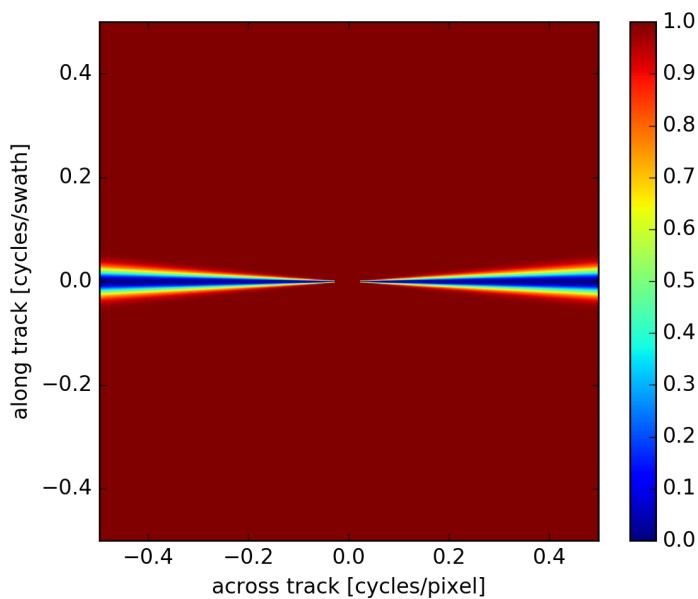

**Figure 8.** Spectral filter $f(\nu_x, \nu_y)$ defined in Eq. 6 to remove CO stripes.

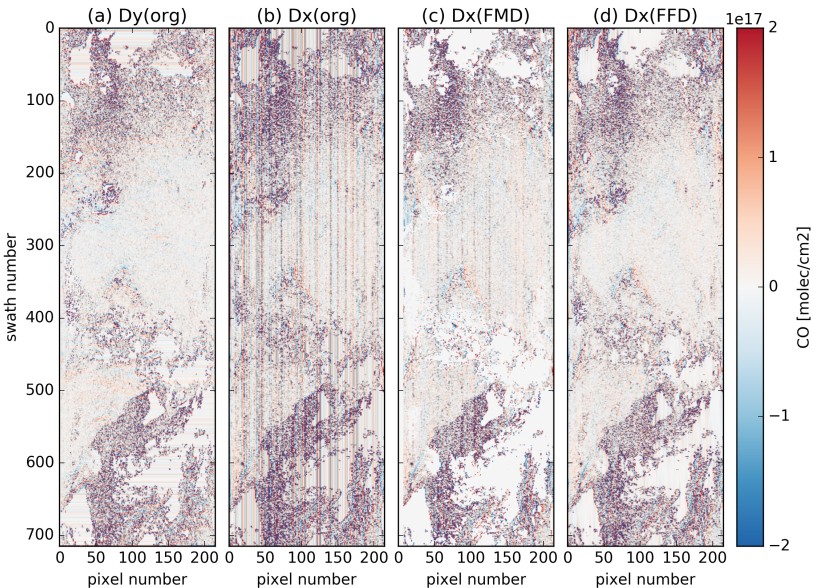

**Figure 9.** CO retrievals of one TROPOMI orbit on 28 July 2018 (partly shown). From left to right: (a) derivative $Dy(\mathbf{d})$ along track of the original data, (b) $Dx(\mathbf{d})$ derivative in cross-track direction of the original data, (c) $Dx(\mathbf{d}_{\mathrm{ds}})$ after FMD destriping , and (d) same as (c) but after FFD destriping.

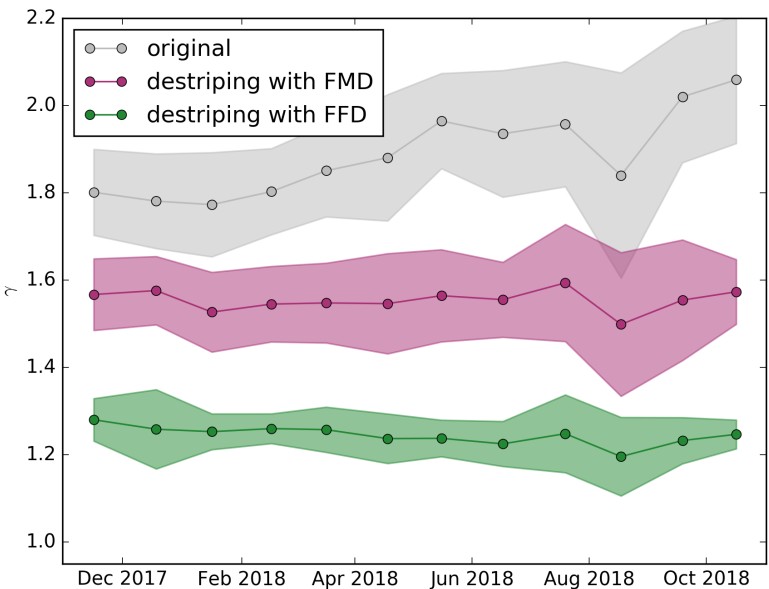

**Figure 10.** The stripiness measure $\gamma$ as defined in Eq. 7 as function of time. (gray) original data, (pink) destriping with FMD approach, (green) destriping with FFD approach. Monthly medians are shown and the shaded area indicates an estimate of the noise (median $\pm$ 84th percentile).

**Table 1.** TROPOMI CO bias with respect to TCCON ($\bar{b}, \bar{\sigma}, \bar{\text{std}}$) and the spectral fit quality ($\bar{\text{rms}}$) in $\text{mol s}^{-1}\,\text{m}^{-2}\,\text{nm}^{-1}\,\text{sr}^{-1}$ as is introduced in Figure 3 for different spectroscopic databases (HITRAN 2008+H2O, SEOM-IAS, HITRAN 2012, and HITRAN 2016). The column 'all' gives the values when the spectroscopic databases are used for all species. The other columns indicate the characteristics when the spectroscopy of only one species is updated. Here, only TROPOMI clear-sky retrievals are considered and no destriping is applied.

| cross-section | statistics | all | CO | $CH_4$ | $H_2O$ | HDO |
|---|---|---|---|---|---|---|
| HITRAN 2008+H2O | $\bar{b}$ | 6.2 | - | - | - | - |
| HITRAN 2008+H2O | $\bar{\sigma}$ | 2.6 | - | - | - | - |
| HITRAN 2008+H2O | $\bar{\text{std}}$ | 3.6 | - | - | - | - |
| HITRAN 2008+H2O | $\bar{\text{rms}}$ | 1.8e-10 | - | - | - | - |
| SEOM-IAS | $\bar{b}$ | 3.4 | 5.8 | 3.3 | 7.6 | 5.2 |
| SEOM-IAS | $\bar{\sigma}$ | 2.0 | 2.5 | 2.1 | 2.6 | 2.6 |
| SEOM-IAS | $\bar{\text{std}}$ | 3.0 | 3.5 | 2.9 | 3.6 | 3.7 |
| SEOM-IAS | $\bar{\text{rms}}$ | 1.5e-10 | 1.8e-10 | 1.5e-10 | 1.7e-10 | 1.8e-10 |
| HITRAN 2012 | $\bar{b}$ | -1.6 | 5.8 | 1.0 | 4.7 | 4.9 |
| HITRAN 2012 | $\bar{\sigma}$ | 1.4 | 2.5 | 1.6 | 2.8 | 2.5 |
| HITRAN 2012 | $\bar{\text{std}}$ | 2.9 | 3.5 | 2.4 | 3.9 | 3.6 |
| HITRAN 2012 | $\bar{\text{rms}}$ | 2.5e-10 | 1.8e-10 | 2.2e-10 | 2.2e-10 | 1.8e-10 |
| HITRAN 2016 | $\bar{b}$ | 0.0 | 5.9 | -0.8 | 8.0 | 5.4 |
| HITRAN 2016 | $\bar{\sigma}$ | 1.8 | 2.5 | 2.0 | 2.4 | 2.6 |
| HITRAN 2016 | $\bar{\text{std}}$ | 2.6 | 3.6 | 2.7 | 3.7 | 3.7 |
| HITRAN 2016 | $\bar{\text{rms}}$ | 1.8e-10 | 1.8e-10 | 1.6e-10 | 2.0e-10 | 1.8e-10 |

**Table 2.** Ground-based TCCON stations used for validation. The latitude and longitude values are given in degrees, the surface elevation in km.

| name | latitude | longitude | altitude | citation |
|---|---|---|---|---|
| Sodankylä | 67.37 | 26.63 | 0.18 | (Kivi et al., 2014; Kivi and Heikkinen, 2016) |
| East Trout Lake | 54.35 | −104.99 | 0.50 | (Wunch et al., 2018) |
| Karlsruhe | 49.10 | 8.44 | 0.11 | (Hase et al., 2015) |
| Orléans | 47.97 | 2.11 | 0.13 | (Warneke et al., 2014) |
| Garmisch | 47.48 | 11.06 | 0.75 | (Sussmann and Rettinger, 2018a) |
| Zugspitze | 47.42 | 10.98 | 2.96 | (Sussmann and Rettinger, 2018b) |
| Park Falls | 45.95 | −90.27 | 0.44 | (Wennberg et al., 2017) |
| Lamont | 36.60 | −97.49 | 0.32 | (Wennberg et al., 2016) |
| Edwards | 34.96 | −117.88 | 0,7 | (Iraci et al., 2016) |
| Pasadena | 34.14 | −118.13 | 0.23 | (Wennberg et al., 2015) |
| Saga | 33.24 | 130.29 | 0.01 | (Kawakami et al., 2014) |
| Izaña | 28.31 | −16.50 | 2.37 | (Blumenstock et al., 2017) |
| Lauder | −45.04 | 169.68 | 0.37 | (Pollard et al., 2019) |