# Peer review of "Improving the TROPOMI CO data product: update of the spectroscopic database and destriping of single orbits"

_Atmospheric Measurement Techniques, 2019_

## Referee Comment (RC1) · Anonymous Referee #1 · 8 Jul 2019

The authors present an update on the TROPOMI CO product, focusing on the evaluation of new spectroscopy and handling of a striping issue. The manuscript is well written and the authors evaluate different spectroscopy databases by comparing retrieved CO against the CAMS-IFS model as well as various ground-based measurements from the TCCON network.

I can recommend the manuscript for publication, however have following minor suggestions/corrections:

Page 5, Line 27: "[...] striping pattern along [...]", should probably read "[...] striping pattern across [...]"

Page 5, Line 30: The authors mention that the striping is not yet understood, however it would be interesting for the reader to know if there is any clue (bad pixels, 'just' calibration?), or if the issue affects any other TROPOMI-derived products. Is it always the same cross-track pixels that show a systematic deviation?

Page 6, Line 5: It would be good to state whether any data manipulation is done for "d" before the 2D Fourier Transformation is applied (filtering, masking, how is missing data treated, etc.)

Figure 4: The std of clear-sky soundings at the Edwards ground station seems "out of order" or considerably higher than cloudy or mixed scenes. I have not found any reference to this in the text, and it seems like an odd outlier that might deserve a sentence or two. Or is this merely due to one overpass towards end of July 2018 (Figure 2)?
* * *

---

## Referee Comment (RC2) · Anonymous Referee #2 · 11 Aug 2019

This paper looks at two improvements to XCO from TropOMI 1) spectroscopic updates, and 2) two types of destriping. Results with different spectroscopy are characterized through comparisons to TCCON. Destriping results are characterized by the "stripiness measure" which quantifies the cross-track vs. along-track variability.

Major comments:

1) This mission requirement is 10% precision and 15% accuracy for single soundings. This work should estimate the precision and accuracy for the different configurations in addition to reporting "rms", "std" and "bias" in Table 1 and the metrics in Figs 2-3.

2) De-striped results should be compared to TCCON to quantify the improvement re-

sulting from de-striping.

Specific comments.

Comment on abstract line 3: To be more clear as to the current TROPOMI configuration, change the wording from "Using HITRAN 2008 spectroscopic data with an updated water vapor spectroscopy, the CO data product is compliant with the mission requirement of 10% precision and 15% accuracy for single soundings." to, "The current TROPOMI is processed using HITRAN 2008 spectroscopic data with an updated water vapor spectroscopy and produces CO products compliant with the mission requirement of 10% precision and 5 15% accuracy for single soundings."

Comment on abstract lines 5-14: The current paper should quantify the precision and accuracy for the different configurations and destriping.

Comment on abstract lines 9-14: The wording says that "HITRAN 2012 ... reduce the bias..." and then later says, "HITRAN 2012 worsens the fitting quality". This is confusing. Does it improve XCO but worsen the spectral fit?

Comment on abstract line 14: The "spectral fitting quality" is not defined (is this the spectral residual?). Ideally report values, rather than it "is worse". Or report about how much worse.

Comment on abstract lines 13: "introduces an artificial bias" Specify the size of bias.

Abstract, line 18 "However, still better quality is achieved..." Comment: "better quality" should be quantified.

Page 3, line 5. "The operational TROPOMI CO processor uses the line lists of HITRAN 2008 (Rothman et al., 2009) ... water vapor". Link this to Table 1, "... water vapor (HITRAN 2008+H2O in Table 1)"

Page 4, line 11. "Table 1 provides the TROPOMI-TCCON mean bias, the standard deviation, and the RMS of the spectral fit residuals when using the current TROPOMI

spectroscopic database, the SEOM-IAS, HITRAN 2012 or HITRAN 2016 data base."
Comment: I do not see the RMS of the spectral fit residuals in Table 1. Table 1 caption
says all values in Table 1 are "CO biases".

Figure 1 caption "Not co-located measurements are marked in gray color." What does
this mean, non-colocated measurements?

Figure 1 caption. The definition of the gray dots is not explained in the caption or text.
There are additionally two sizes of gray dots.

Figure 1 caption. The blue and gray are hard to distinguish, either make the gray or
blue darker.

Figure 3 caption "mean bias (TROPOMI - TCCON) between co-located daily mean
XCO values (see Fig. 1, 2) of TROPOMI and TCCON". Is this using the pink, gray, or
both types of dots from Fig. 2?

Figure 3-4. Define sigma-bar in (a), std-bar in (b) and rms-bar in (c).

Table 1. "Table 1. TROPOMI CO bias with respect to TCCON (bias, std, and rms)".
This is not de-striped, correct? Bias, std, and rms need to be defined. The text says
this is the spectral rms, but Table 1 states this are all "XCO biases". Do these relate to
b-bar, etc., from Fig. 2? Sigma-bar from Fig 2 needs to be included in this table as this
is part of the systematic error.

A table needs to be shown with estimates of the precision and accuracy to link to the
mission requirements. E.g. subtracting out the mean bias and combining with the bias
variability and std-bar to estimate the accuracy, and using rms-bar for precision (if I
understand these terms.)

Add comparisons to TCCON after destriping with the two types of destriping into a
table, either to Table 1 or an additional table. Although the FFD destriping method
shown in Fig 7 looks better comparisons to TCCON are needed to quantify if destriping
improves precision and accuracy.

---

## Referee Comment (RC3) · Anonymous Referee #3 · 20 Aug 2019

This paper discusses improvements to the XCO product from TROPOMI, specifically the use of updated spectroscopic databases (evaluated through comparisons with TC-CON and the CAMS-IFS model) and two types of destriping methods.

General Comment:

The authors should be careful to exactly define what is meant by certain terms when they are introduced in the text. For instance what is implied by the average bias and standard deviation of the bias? Are all collocated data pairs averaged regardless of their station origin (implying a stronger impact on the average bias for TCCON stations that have many collocations (for instance East Trout Lake) versus stations that have

few (Edwards)) or is it the average of the individual station biases? From the caption in Fig 3 I assume the latter but this should be mentioned in the main text. Similarly for 'the station-to-station variability of the bias' (again I assume std of the bias over different stations based on the figures)

To evaluate the significance of a bias, the standard error or confidence interval is a far more useful metric than the standard deviation of the bias. Particularly when evaluation different products.

Specific comments:

Concerning the collocation criteria with TCCON: In this study data are collocated within a 50km radius and within the same day. Given that average wind speeds in the free troposphere can quickly reach values of 20 to 30 km/hour, a 2 hour collocation window would be a better match for the chosen spatial collocation radius.

Fig 5: The caption mentions that it is like Fig 3. Yet the rms plot is replaced by a number of coincidences plot. This should be mentioned.

---

## Author Comment (AC1) · 9 Sep 2019

We would like to thank reviewer 1, 2, 3 for the constructive comments that aided us to improve our manuscript. In this post we provide our replies to the reviewer's comments. We provide a revised version of the manuscript, in which all changes are highlighted. Revised and added text is provided in blue. In our replies to the comment we provide line numbers, page numbers and figure numbers of the old version of the manuscript.

Please also note the supplement to this comment:

[Figure]

https://www.atmos-meas-tech-discuss.net/amt-2019-241/amt-2019-241-AC1-supplement.pdf

[Figure]

**Supplement:**

**author comments on the manuscript "Improving the TROPOMI CO data product: update of the spectroscopic database and destriping of single orbits", reviewer 1**

We would like to thank the reviewer for the constructive comments that aided us to improve our manuscript. In this document we provide our replies to the reviewer's comments. The original comments made by the reviewer are numbered and typeset in italic and bold face font. Following every comment we give our reply. Here line numbers, page numbers and figure numbers refer to the original version of the manuscript, if not stated differently. Additionally, the revised version of the manuscript is added.

1. *Page 5, Line 27: "[...] striping pattern along [...]", should probably read "[...] striping pattern across [...]"*

   **not adjusted**

   We see stripes of the CO concentration in flight direction and not across the flight direction. Hence, we think that our description is correct.

2. *Page 5, Line 30: The authors mention that the striping is not yet understood, however it would be interesting for the reader to know if there is any clue (bad pixels, just calibration?), or if the issue affects any other TROPOMI-derived products. Is it always the same cross-track pixels that show a systematic deviation?*

   **adjusted**

   We changed the sentence p5,l30 from:
   "The origin of the stripy pattern is not yet understood and . . . "
   to
   "The origin of the stripy pattern is not yet understood and is changing with time from orbit to orbit. The TROPOMI level 1 team is optimizing the Calibration Key Data (CKD) to reduce the effect in future."

3. *Page 6, Line 5: It would be good to state whether any data manipulation is done for "d" before the 2D Fourier Transformation is applied (filtering, masking, how is missing data treated, etc.)*

   **adjusted**

   We added the following sentence p6,l8:
   "Before this transformation the missing data in **d** was replaced by the median value of the corresponding swath and additionally a fixed strip pattern was added to the interpolated missing values deploying the FMD method. "

4. *Figure 4: The std of clear-sky soundings at the Edwards ground station seems "out of order" or considerably higher than cloudy or mixed scenes. I have not found any reference to this in the text, and it seems like an odd outlier that might deserve a sentence or two. Or is this merely due to one overpass towards end of July 2018 (Figure 2)?*

   **adjusted** Yes, it is due to an outlier. We updated our complete analysis that is now based on robust statistics using the percentile difference $\delta P$ (for definition see eq. 1), the data scatter for Edwards is in agreement with that of the other stations. (see also the discussion with referee 2).

**Additional changes**

We added an additional co-author to the document "Thorsten Warneke". The affiliation of Dietrich Feist was changed and an acknowledgement for the German TCCON stations was added.

[revised manuscript text omitted]

We would like to thank the reviewer for the constructive comments that asked us to improve our manuscript. In this document we provide our replies to the reviewer's comments. The original comments made by the reviewer are numbered and typeset in italic and bold face font. Following every comment we give our reply. Here line numbers, page numbers and figure numbers refer to the original version of the manuscript, if not stated differently. Additionally, the revised version of the manuscript is added.

1. *Page 5, Line 27: "[...] striping pattern along [...]", should probably read "[...] striping pattern across [...]"*

   **not adjusted**

   We see stripes of the CO concentration in flight direction and not across the flight direction. Hence, we think that our description is correct.

2. *Page 5, Line 30: The authors mention that the striping is not yet understood, however it would be interesting for the reader to know if there is any clue (bad pixels, just calibration?), or if the issue affects any other TROPOMI-derived products. Is it always the same cross-track pixels that show a systematic deviation?*

   **adjusted**

   We changed the sentence p5,l30 from:

   "The origin of the stripy pattern is not yet understood and ..."

   to

   "The origin of the stripy pattern is not yet understood and is changing with time from orbit to orbit. The TROPOMI level 1 team is optimizing the Calibration Key Data (CKD) to reduce the effect in future."

3. *Page 6, Line 5: It would be good to state whether any data manipulation is done for "d" before the 2D Fourier Transformation is applied (filtering, masking, how is missing data treated, etc.)*

   **adjusted**

   We added the following sentence p6,l8:

   "Before this transformation the missing data in d was replaced by the median value of the corresponding swath and additionally a fixed strip pattern was added to the interpolated missing values deploying the FMD method."

4. *Figure 4: The std of clear-sky soundings at the Edwards ground station seems "out of order" or considerably higher than cloudy or mixed scenes. I have not found any reference to this in the text, and it seems like an odd outlier that might deserve a sentence or two. Or is this merely due to one overpass towards end of July 2018 (Figure 2)?*

   **adjusted** Yes, it is due to an outlier. We updated our complete analysis that is now based on robust statistics using the percentile difference $\delta P$ (for definition see eq. 1), the data scatter for Edwards is in agreement with that of the other stations. (see also the discussion with referee 2).

**Additional changes**

We added an additional co-author to the document "Thorsten Warneke". The affiliation of Dietrich Feist was changed and an acknowledgement for the German TCCON stations was added.

[revised manuscript text omitted]

---

## Author Comment (AC2) · 9 Sep 2019

We would like to thank reviewer 1, 2, and 3 for the constructive comments that aided us to improve our manuscript. In this post we provide our replies to the reviewer's comments. We provide a revised version of the manuscript, in which all changes are highlighted. Revised and added text is provided in blue. In our replies to the comment we provide line numbers, page numbers and figure numbers of the old version of the manuscript.

Please also note the supplement to this comment:

[Figure]

https://www.atmos-meas-tech-discuss.net/amt-2019-241/amt-2019-241-AC2-supplement.pdf

[Figure]

**Supplement:**

**author comments on the manuscript "Improving the TROPOMI CO data product: update of the spectroscopic database and destriping of single orbits", reviewer 2**

We would like to thank the reviewer for the constructive comments that aided us to improve our manuscript. In this document we provide our replies to the reviewer's comments. The original comments made by the reviewer are numbered and typeset in italic and bold face font. Following every comment we give our reply. Here line numbers, page numbers and figure numbers refer to the original version of the manuscript, if not stated differently. Additionally, the revised version of the manuscript is added.

**Major comments**

1. *This mission requirement is 10% precision and 15% accuracy for single soundings. This work should estimate the precision and accuracy for the different configurations in addition to reporting "rms", "std" and "bias" in Table 1 and the metrics in Figs 2-3.*

   **not adjusted**

   We understand the reviewers comments such that to evaluate the precision and accuracy requirements of the mission also a direct estimate of these quantities is desired. However, the basic quantity that we can evaluate is the difference between the satellite observation and a ground truth for circumstances which changes from observation to observation. So, the data set does not include measurements of the same measurement and the observed difference is a result of measurement precision, representation errors, errors in the ground truth and measurement biases, latter varying on different temporal and spatial scales. Hence, strictly speaking quantities like precision and accuracy cannot be derived from these data sets but related quantities to describe the data quality can be provided.

   In the paper, we follow an error characterization adopted from corresponding validation studies of GOSAT and OCO-2 (Cogan et al. (2012), Wu et al. (2018)), which was applied to TROPOMI data already Borsdorff et al. (2018), It analysis the bias and the scatter of difference time series of collocated TCCON and satellite observations. Here, the mean bias indicates the trueness/accuracy averaged over the period of the time series. The percentile difference $\delta P$, used in the revised version of the manuscript is a measure of the scatter of the differences between TROPOMI and TCCON and combines precision, pseudo-noise and other biases. To our opinion, it is not possible to isolate the contribution of precision. However, we can consider $\delta P$ as a upper boundary of the CO precision. In accordance with previous validation studies, we prefer to follow this approach. To prevent misinterpretations, we added the following sentence the manuscript p2,l20:
   " Here, the bias between TROPOMI CO and the TCCON measurements was used to estimate the product accuracy and the scatter in the difference between both measurements indicated an upper boundary for the accuracy and precision of the TROPOMI instrument."

2. *De-striped results should be compared to TCCON to quantify the improvement re- sulting from de-striping.*

   **adjusted** We already validated the de-striped TROPOMI CO data with TCCON measurements in the submitted version of the manuscript (see p7, l7-12). We found that the validation approach is not sensitive to striping patterns in the data product. Therefore, we developed a different verification approach as presented in Sec. 3.2 of the manuscript. We revisited this conclusion deploying the more robust statistics against outliers used for the new manuscript, which confirmed our previous finding. To make this more clear we add the following paragraph to the conclusions:
   " For both destriping methods, we found that the TCCON validation (bias, station-to-station variability of the bias, and scatter of the bias) does not significantly change. For the TCCON validation daily averages in a collocation radius of 50 km were calculated. We found that on this scale, the impact of stripes on single orbit data can be neglected."

**Specific comments**

1. *Comment on abstract line 3: To be more clear as to the current TROPOMI configuration, change the wording from "Using HITRAN 2008 spectroscopic data with an updated water vapor spectroscopy, the CO data product is compliant with the mission requirement of 10% precision and 15% accuracy for single soundings." to, "The current TROPOMI is*

*processed using HITRAN 2008 spectroscopic data with an updated water vapor spectroscopy and produces CO products compliant with the mission requirement of 10% precision and 15% accuracy for single soundings."*

**adjusted**

We follow the suggestion of the reviewer and changed the sentence p1,l3 from:
" Using HITRAN 2008 spectroscopic data with an updated water vapor spectroscopy, the CO data product is compliant with the mission requirement of 10 % precision and 15 % accuracy for single soundings. "
to
" The current TROPOMI CO processing uses the HITRAN 2008 spectroscopic data with an updated water vapor spectroscopy and produces a CO data product compliant with the mission requirement of 10% precision and 15% accuracy for single soundings. "

2. *Comment on abstract lines 5-14: The current paper should quantify the precision and accuracy for the different configurations and destriping.*
   **not adjusted** Please see our answer to major comment 1 of this review.

3. *Comment on abstract lines 9-14: The wording says that "HITRAN 2012 ... reduce the bias..." and then later says, "HITRAN 2012 worsens the fitting quality". This is confusing. Does it improve XCO but worsen the spectral fit?*
   **adjusted**

   Indeed, HITRAN 2012 reduced the overall bias with respect to TCCON, it introduces an artificial bias in the tropics between TROPOMI and CAMS and yields the worst fitting quality of all tested cross-sections (please see table 1) To make this more clear we change the following sentence in the abstract p1,l12 from:

   " Here, HITRAN 2012 worsens the fitting quality and furthermore introduces an artificial bias to the TROPOMI CO data product in the tropics caused by the $H_2O$ spectroscopic data." to

   " HITRAN 2012 shows the worst fit quality (rms=2.5e-10 $mol\,s^{-1}\,m^{-2}\,nm^{-1}\,sr^{-1}$ ) of the tested cross-sections and furthermore introduces an artificial bias of about -1.5e17 molec/$cm^2$ between TROPOMI CO and the CAMS-IFS model in the tropics caused by the $H_2O$ spectroscopic data. "

4. *Comment on abstract line 14: The "spectral fitting quality" is not defined (is this the spectral residual?). Ideally report values, rather than it "is worse". Or report about how much worse.*
   **adjusted**

   We change the sentence p1, l11 from: "SEOM-IAS achieves the best spectral fitting quality and reduces the bias between TROPOMI and TCCON ..."
   to " SEOM-IAS achieves the best spectral fit quality (root-mean-squared (rms) differences between the simulated and measured spectrum) of 1.5e-10 $mol\,s^{-1}\,m^{-2}\,nm^{-1}\,sr^{-1}$ and reduces the bias between TROPOMI and TCCON ..."

   we added the value of the fit quality also to p7,l28 and p7,l26.

5. *Comment on abstract lines 13: "introduces an artificial bias" Specify the size of bias.*
   **adjusted**

   We changed the sentence p1,l13 from:
   "...introduces an artificial bias " to
   "...introduces an artificial bias of about -1.5e17 molec/$cm^2$"

   The values of the bias is also added to the manuscript p7,29 and p5,l15.

6. *Abstract, line 18 "However, still better quality is achieved..." Comment: "better quality" should be quantified.*
   **adjusted** We changed the sentence p1,l18 from:
   "However, still better quality is achieved by a Fourier analysis and filtering ..."
   to " However, the destriping can be further improved achieving $\gamma = 1.2$ deploying a Fourier analysis and filtering ..."

   Furthermore we changed the sentence p1,l16 from

   "A destriping mask calculated per orbit by median filtering of the data in the cross-track direction significantly improves the data quality." to

" A destriping mask calculated per orbit by median filtering of the data in the cross-track direction significantly reduced the stripe pattern from $\gamma = 2.1$ to $\gamma = 1.6$. "

7. **Page 3, line 5. "The operational TROPOMI CO processor uses the line lists of HITRAN 2008 (Rothman et al., 2009) ... water vapor". Link this to Table 1, "... water vapor (HITRAN 2008+H2O in Table 1)"**
   **adjusted** We changed the sentence p3,l5 from:
   "...water vapor." to
   " water vapor (HITRAN 2008+H2O in Table 1) "

8. **Page 4, line 11. "Table 1 provides the TROPOMI-TCCON mean bias, the standard deviation, and the RMS of the spectral fit residuals when using the current TROPOMI spectroscopic database, the SEOM-IAS, HITRAN 2012 or HITRAN 2016 data base." Comment: I do not see the RMS of the spectral fit residuals in Table 1. Table 1 caption says all values in Table 1 are "CO biases".**
   **adjusted**

   We added the rms of the spectral fit quality to the Table 1 and changed its caption from:

   " TROPOMI CO bias with respect to TCCON (bias, std, and rms) for different spectroscopic databases (HITRAN 2008+H2O, SEOM-IAS, HITRAN 2012, and HITRAN 2016). The column 'all' gives the bias when the spectroscopic databases are used for all species. The other columns indicate the bias when the spectroscopy of only one species is updated. Here, only TROPOMI clear-sky retrievals are considered. " to

   " TROPOMI CO bias with respect to TCCON ($\bar{b}$, $\bar{\sigma}$, $\bar{std}$) and the spectral fit quality ($\bar{rms}$) in $\mathrm{mol\,s^{-1}\,m^{-2}\,nm^{-1}\,sr^{-1}}$ as is introduced in Figure 3 for different spectroscopic databases (HITRAN 2008+H2O, SEOM-IAS, HITRAN 2012, and HITRAN 2016). The column 'all' gives the values when the spectroscopic databases are used for all species. The other columns indicate the characteristics when the spectroscopy of only one species is updated. Here, only TROPOMI clear-sky retrievals are considered and no destriping is applied. "

9. **Figure 1 caption "Not co-located measurements are marked in gray color." What does this mean, non-colocated measurements? Figure 1 caption. The definition of the gray dots is not explained in the caption or text. There are additionally two sizes of gray dots.**
   **adjusted** We changed the sentence in the caption of Figure 1 from:
   "Not co-located measurements are marked in grey color." to
   " Measurements of both datasets that could not be paired are marked as grey dots (big=TROPOMI, small=TCCON) and are not used in this study. "

10. **Figure 1 caption. The blue and gray are hard to distinguish, either make the gray or blue darker.**
    **adjusted**

    We changed the colors in Figure 1 accordingly.

11. **Figure 3 caption "mean bias (TROPOMI - TCCON) between co-located daily mean XCO values (see Fig. 1, 2) of TROPOMI and TCCON". Is this using the pink, gray, or both types of dots from Fig. 2?**
    **adjusted**

    We changed the Figure caption from:
    "between co-located daily mean XCO values (see Fig. 1,2) " to "between co-located daily mean XCO values (see blue and pink dots in Fig. 1 and 2) "

12. **Figure 3-4. Define sigma-bar in (a), std-bar in (b) and rms-bar in (c).**
    **adjusted**

    We changed the caption of figure from :
    " $\bar{b}$ is the global mean bias (average of all station biases) and $\bar{\sigma}$ is the bias standard deviation. $\bar{std}$ is the average of all standard deviations and $\bar{rms}$ the average rms of coincident daily mean pairs from TROPOMI and TCCON." to
    " (a) median bias $b_j$ (TROPOMI-TCCON) for different TCCON sites between co-located daily mean XCO values of TROPOMI and TCCON (see blue and pink dots in Fig. 1, 2) The global mean bias $\bar{b}$ and the correspsonding standard derivation $\bar{\sigma}$ as defined in Eq. (2) and (3), (b) the scatter $\delta P_j$ of the biases as defined in Eq. (1) with its global mean $\bar{\delta P}$ and (c) the median root-mean-square (rms) of the spectral

fit residuals of the individual retrievals per station and its global mean r$\bar{\text{m}}$s in $\mathrm{mol\,s^{-1}\,m^{-2}\,nm^{-1}\,sr^{-1}}$. The figure shows TROPOMI retrievals under clear-sky (yellow), cloudy-sky (blue) and the combination of both (pink). No destriping is applied to the TROPOMI data. The retrieval deploys the spectroscopic database HITRAN 2016 for all absorbers. ”

13. ***Table 1. ”Table 1. TROPOMI CO bias with respect to TCCON (bias, std, and rms)”. This is not de-striped, correct? Bias, std, and rms need to be defined. The text says this is the spectral rms, but Table 1 states this are all ”XCO biases”. Do these relate to b-bar, etc., from Fig. 2? Sigma-bar from Fig 2 needs to be included in this table as this is part of the systematic error.***
    **adjusted**

    Table 1 gives the results for data without any bias correction. All diagnostic tools are now defined at p3. l28. We changed the caption from:
    “ TROPOMI CO bias with respect to TCCON (bias, std) and the spectral fit quality (rms) for different spectroscopic databases (HITRAN 2008+H2O, SEOM-IAS, HITRAN 2012, and HITRAN 2016). The column 'all' gives the values when the spectroscopic databases are used for all species. The other columns indicate the characteristics when the spectroscopy of only one species is updated. Here, only TROPOMI clear-sky retrievals are considered. ” to
    “ TROPOMI CO bias with respect to TCCON ($\bar{b}$, $\bar{\sigma}$, s$\bar{\text{t}}$d) and the spectral fit quality (r$\bar{\text{m}}$s) in $\mathrm{mol\,s^{-1}\,m^{-2}\,nm^{-1}\,sr^{-1}}$ as is introduced in Figure 3 for different spectroscopic databases (HITRAN 2008+H2O, SEOM-IAS, HITRAN 2012, and HITRAN 2016). The column 'all' gives the values when the spectroscopic databases are used for all species. The other columns indicate the characteristics when the spectroscopy of only one species is updated. Here, only TROPOMI clear-sky retrievals are considered and no destriping is applied. ”

    Additionally we added $\bar{\sigma}$ to the table and renamed the rows accordingly.

14. ***A table needs to be shown with estimates of the precision and accuracy to link to the mission requirements. E.g. subtracting out the mean bias and combining with the bias variability and std-bar to estimate the accuracy, and using rms-bar for precision (if I understand these terms.)***
    **not adjusted**

    rms-bar is the spectral fit quality and no bias estimate. It was not well defined in the old manuscript and let to this confusion. Following the advice of the referee this is defined now. For the discussion about the estimation of precision and accuracy please see our response to the major comment 1 of this referee.

15. ***Add comparisons to TCCON after destriping with the two types of destriping into a table, either to Table 1 or an additional table. Although the FFD destriping method shown in Fig 7 looks better comparisons to TCCON are needed to quantify if destriping improves precision and accuracy.***
    **not adjusted** Please see our answer to the major comment 2 of this referee.

**not adjusted**

We understand the reviewers comments such that to evaluate the precision and accuracy requirements of the mission also a direct estimate of these quantities is desired. However, the basic quantity that we can evaluate is the difference between the satellite observation and a ground truth for circumstances which changes from observation to observation. So, the data set does not include measurements of the same measurement and the observed difference is a result of measurement precision, representation errors, errors in the ground truth and measurement biases, latter varying on different temporal and spatial scales. Hence, strictly speaking quantities like precision and accuracy cannot be derived from these data sets but related quantities to describe the data quality can be provided.

In the paper, we follow an error characterization adopted from corresponding validation studies of GOSAT and OCO-2 (Cogan et al. (2012), Wu et al. (2018)), which was applied to TROPOMI data already Borsdorff et al. (2018). It analysis the bias and the scatter of difference time series of collocated TCCON and satellite observations. Here, the mean bias indicates the trueness/accuracy averaged over the period of the time series. The percentile difference $\delta P$, used in the revised version of the manuscript is a measure of the scatter of the differences between TROPOMI and TCCON and combines precision, pseudo-noise and other biases. To our opinion, it is not possible to isolate the contribution of precision. However, we can consider $\delta P$ as a upper boundary of the CO precision. To prevent misinterpretations, we added the following sentence the manuscript p2,l20:

"Here, the bias between TROPOMI CO and the TCCON measurements was used to estimate the product accuracy and the scatter in the difference between both measurements indicated an upper boundary for the accuracy and precision of the TROPOMI instrument."

2. *De-striped results should be compared to TCCON to quantify the improvement re- sulting from de-striping.*

**adjusted** We already validated the de-striped TROPOMI CO data with TCCON measurements in the submitted version of the manuscript (see p7, l7-12). We found that the validation approach is not sensitive to striping patterns in the data product. Therefore, we developed a different verification approach as presented in Sec. 3.2 of the manuscript. We revisited this conclusion deploying the more robust statistics against outliers used for the new manuscript, which confirmed our previous finding. To make this more clear we add the following paragraph to the conclusions

"For both destriping methods, we found that the TCCON validation (bias, station-to-station variability of the bias, and scatter of the bias) does not significantly change. For the TCCON validation daily averages in a collocation radius of 50 km were calculated. We found that on this scale, the impact of stripes on single orbit data can be neglected."

**Specific comments**

1. *Comment on abstract line 3: To be more clear as to the current TROPOMI configura- tion, change the wording from "Using HITRAN 2008 spectroscopic data with an updated water vapor spectroscopy, the CO data product is compliant with the mission requirement of 10% precision and 15% accuracy for single soundings." to, "The current TROPOMI is*

*processed using HITRAN 2008 spectroscopic data with an updated water vapor spectroscopy and produces CO products compliant with the mission requirement of 10% precision and 15% accuracy for single soundings."*

**adjusted**

We follow the suggestion of the reviewer and changed the sentence p1.13 from:

" Using HITRAN 2008 spectroscopic data with an updated water vapor spectroscopy, the CO data product is compliant with the mission requirement of 10% precision and 15% accuracy for single soundings. "

to

" The current TROPOMI CO processing uses the HITRAN 2008 spectroscopic data with an updated water vapor spectroscopy and produces a CO data product compliant with the mission requirement of 10% precision and 15% accuracy for single soundings. "

2. *Comment on abstract lines 5-14: The current paper should quantify the precision and accuracy for the different configurations and destriping.*

not adjusted Please see our answer to major comment 1 of this review.

3. *Comment on abstract lines 9-14: The wording says that "HITRAN 2012 ... reduce the bias..." and then later says, "HITRAN 2012 worsens the fitting quality". This is confusing. Does it improve XCO but worsen the spectral fit?*

**adjusted**

Indeed, HITRAN 2012 reduced the overall bias with respect to TCCON, it introduces an artificial bias in the tropics between TROPOMI and CAMS and yields the worst fitting quality of all tested cross-sections (please see table 1) To make this more clear we change the following sentence in the abstract p1.112 from:

" Here, HITRAN 2012 worsens the fitting quality and furthermore introduces an artificial bias to the TROPOMI CO data product in the tropics caused by the $H_2O$ spectroscopic data." to

" HITRAN 2012 shows the worst fit quality (rms=2.5e-10 mol s$^{-1}$ m$^{-2}$ nm$^{-1}$ sr$^{-1}$) of the tested cross-sections and furthermore introduces an artificial bias of about -1.5e17 molec/cm$^2$ between TROPOMI CO and the CAMS-IFS model in the tropics caused by the $H_2O$ spectroscopic data. "

4. *Comment on abstract line 14: The "spectral fitting quality" is not defined (is this the spectral residual?). Ideally report values, rather than it "is worse". Or report about how much worse.*

**adjusted**

We change the sentence p1.111 from: "SEOM-IAS achieves the best spectral fitting quality and reduces the bias between TROPOMI and TCCON ... "

to "SEOM-IAS achieves the best spectral fit quality (root-mean-squared (rms) differences between the simulated and measured spectrum) of 1.5e-10 mol s$^{-1}$ m$^{-2}$ nm$^{-1}$ sr$^{-1}$ and reduces the bias between TROPOMI and TCCON ... "

we added the value of the fit quality also to p7.128 and p7.126.

5. *Comment on abstract lines 13: "introduces an artificial bias" Specify the size of bias.*

**adjusted**

We changed the sentence p1.113 from:
" ...introduces an artificial bias " to
" ...introduces an artificial bias of about -1.5e17 molec/cm$^2$

The values of the bias is also added to the manuscript p7.29 and p5.115.

6. *Abstract, line 18 "However, still better quality is achieved..." Comment: "better quality" should be quantified.*

**adjusted** We changed the sentence p1.118 from:

"However, still better quality is achieved by a Fourier analysis and filtering ... "

to " However, the destriping can be further improved achieving $\gamma = 1.2$ deploying a Fourier analysis and filtering ... "

Furthermore we changed the sentence p1.116 from

"A destriping mask calculated per orbit by median filtering of the data in the cross-track direction significantly improves the data quality." to

" A destriping mask calculated per orbit by median filtering of the data in the cross-track direction significantly reduced the stripe pattern from $\gamma = 2.1$ to $\gamma = 1.6$. "

7. **Page 3, line 5. "The operational TROPOMI CO processor uses the line lists of HITRAN 2008 (Rothman et al., 2009) ... water vapor". Link this to Table 1, "... water vapor (HITRAN 2008+H2O in Table 1)"**
   adjusted We changed the sentence p3,l5 from:

   "... water vapor." to

   " water vapor (HITRAN 2008+H2O in Table 1) "

8. **Page 4, line 11. "Table 1 provides the TROPOMI-TCCON mean bias, the standard deviation, and the RMS of the spectral fit residuals when using the current TROPOMI spectroscopic database, the SEOM-IAS, HITRAN 2012 or HITRAN 2016 data base." Comment: I do not see the RMS of the spectral fit residuals in Table 1. Table 1 caption says all values in Table 1 are "CO biases".**
   adjusted

   We added the rms of the spectral fit quality to the Table 1 and changed its caption from:

   " TROPOMI CO bias with respect to TCCON (bias, std, and rms) for different spectroscopic databases (HITRAN 2008+H2O, SEOM-IAS, HITRAN 2012, and HITRAN 2016). The column 'all' gives the bias when the spectroscopic databases are used for all species. The other columns indicate the bias when the spectroscopy of only one species is updated. Here, only TROPOMI clear-sky retrievals are considered. " to

   " TROPOMI CO bias with respect to TCCON ($\bar{b}$, $\bar{\sigma}$, std) and the spectral fit quality (rms) in mols$^{-1}$ m$^{-2}$ nm$^{-1}$ sr$^{-1}$ as is introduced in Figure 3 for different spectroscopic databases (HITRAN 2008+H2O, SEOM-IAS, HITRAN 2012, and HITRAN 2016). The column 'all' gives the values when the spectroscopic databases are used for all species. The other columns indicate the characteristics when the spectroscopy of only one species is updated. Here, only TROPOMI clear-sky retrievals are considered and no destriping is applied. "

9. **Figure 1 caption "Not co-located measurements are marked in gray color." What does this mean, non-colocated measurements? Figure 1 caption. The definition of the gray dots is not explained in the caption or text. There are additionally two sizes of gray dots.**
   adjusted We changed the sentence in the caption of Figure 1 from:

   "Not co-located measurements are marked in grey color." to

   " Measurements of both datasets that could not be paired are marked as grey dots (big=TROPOMI, small=TCCON) and are not used in this study. "

10. **Figure 1 caption. The blue and gray are hard to distinguish, either make the gray or blue darker.**
    adjusted

    We changed the colors in Figure 1 accordingly.

11. **Figure 3 caption "mean bias (TROPOMI - TCCON) between co-located daily mean XCO values (see Fig. 1, 2) of TROPOMI and TCCON". Is this using the pink, gray, or both types of dots from Fig. 2?**
    adjusted

    We changed the Figure caption from:

    "between co-located daily mean XCO values (see Fig. 1,2) " to "between co-located daily mean XCO values (see blue and pink dots in Fig. 1 and 2) "

12. **Figure 3-4. Define sigma-bar in (a), std-bar in (b) and rms-bar in (c).**
    adjusted

    We changed the caption of figure from :

    " $\bar{b}$ is the global mean bias (average of all station biases) and $\bar{\sigma}$ is the bias standard deviation. std is the average of all standard deviations and rms the average rms of coincident daily mean pairs from TROPOMI and TCCON." to

    " (a) median bias $b_j$ (TROPOMI-TCCON) for different TCCON sites between co-located daily mean XCO values of TROPOMI and TCCON (see blue and pink dots in Fig. 1, 2) The global mean bias $\bar{b}$ and the corresponding standard derivation $\bar{\sigma}$ as defined in Eq. (2) and (3), (b) the scatter $\delta P_j$ of the biases as defined in Eq. (1) with its global mean $\delta P$ and (c) the median root-mean-square (rms) of the spectral

fit residuals of the individual retrievals per station and its global mean rms in mols$^{-1}$ m$^{-2}$ nm$^{-1}$ sr$^{-1}$. The figure shows TROPOMI retrievals under clear-sky (yellow), cloudy-sky (blue) and the combination of both (pink). No destriping is applied to the TROPOMI data. The retrieval deploys the spectroscopic database HITRAN 2016 for all absorbers."

13. *"Table 1. TROPOMI CO bias with respect to TCCON (bias, std, and rms)". This is not de-striped, correct? Bias, std, and rms need to be defined. The text says this is the spectral rms, but Table 1 states this are all "XCO biases". Do these relate to b-bar, etc., from Fig. 2? Sigma-bar from Fig 2 needs to be included in this table as this is part of the systematic error.*

adjusted

Table 1 gives the results for data without any bias correction. All diagnostic tools are now defined at p3. 28. We changed the caption from:

"TROPOMI CO bias with respect to TCCON (bias, std) and the spectral fit quality (rms) for different spectroscopic databases (HITRAN 2008–H2O, SEOM-IAS, HITRAN 2012, and HITRAN 2016). The column 'all' gives the values when the spectroscopic databases are used for all species. The other columns indicate the characteristics when the spectroscopy of only one species is updated. Here, only TROPOMI clear-sky retrievals are considered. " to

"TROPOMI CO bias with respect to TCCON ($\bar{b}$, $\bar{\sigma}$, std) and the spectral fit quality (rms) in mols$^{-1}$ m$^{-2}$ nm$^{-1}$ sr$^{-1}$ as is introduced in Figure 3 for different spectroscopic databases (HITRAN 2008–H2O, SEOM-IAS, HITRAN 2012, and HITRAN 2016). The column 'all' gives the values when the spectroscopic databases are used for all species. The other columns indicate the characteristics when the spectroscopy of only one species is updated. Here, only TROPOMI clear-sky retrievals are considered and no destriping is applied. "

Additionally we added $\bar{\sigma}$ to the table and renamed the rows accordingly.

14. *A table needs to be shown with estimates of the precision and accuracy to link to the mission requirements. E.g. subtracting out the mean bias and combining with the bias variability and std-bar to estimate the accuracy, and using rms-bar for precision (if I understand these terms.)*

not adjusted

rms-bar is the spectral fit quality and no bias estimate. It was not well defined in the old manuscript and let to this confusion. Following the advice of the referee this is defined now. For the discussion about the estimation of precision and accuracy please see our response to the major comment 1 of this referee.

15. *Add comparisons to TCCON after destriping with the two types of destriping into a table, either to Table 1 or an additional table. Although the FFD destriping method shown in Fig 7 looks better comparisons to TCCON are needed to quantify if destriping improves precision and accuracy.*

not adjusted Please see our answer to the major comment 2 of this referee.

[revised manuscript text omitted]

---

## Author Comment (AC3) · 9 Sep 2019

We would like to thank reviewer 1, 2, and 3 for the constructive comments that aided us to improve our manuscript. In this post we provide our replies to the reviewer's comments. We provide a revised version of the manuscript, in which all changes are highlighted. Revised and added text is provided in blue. In our replies to the comment we provide line numbers, page numbers and figure numbers of the old version of the manuscript.

Please also note the supplement to this comment:

[Figure]

https://www.atmos-meas-tech-discuss.net/amt-2019-241/amt-2019-241-AC3-supplement.pdf

[Figure]

**Supplement:**

**author comments on the manuscript "Improving the TROPOMI CO data product: update of the spectroscopic database and destriping of single orbits", reviewer 3**

We would like to thank the reviewer for the constructive comments that aided us to improve our manuscript. In this document we provide our replies to the reviewer's comments. The original comments made by the reviewer are numbered and typeset in italic and bold face font. Following every comment we give our reply. Here line numbers, page numbers and figure numbers refer to the original version of the manuscript, if not stated differently. Additionally, the revised version of the manuscript is added.

**Major comments**

1. ***The authors should be careful to exactly define what is meant by certain terms when they are introduced in the text. For instance what is implied by the average bias and standard deviation of the bias? Are all collocated data pairs averaged regardless of their station origin (implying a stronger impact on the average bias for TCCON stations that have many collocations (for instance East Trout Lake) versus stations that have few (Edwards)) or is it the average of the individual station biases? From the caption in Fig 3 I assume the latter but this should be mentioned in the main text.***

2. ***Similarly for the station-to-station variability of the bias (again I assume std of the bias over different stations based on the figures) To evaluate the significance of a bias, the standard error or confidence interval is a far more useful metric than the standard deviation of the bias. Particularly when evaluation different products.***

   **adjusted**

   We agree with both comments of the reviewer. Therefore, we adapted our study such that our statistical analysis is based on the median bias and the percentile difference $\delta P$ (see definition below) to quantify the scatter in the biases per station. The overall conclusions of our study remains the same. Here, the data scatter $\delta P$ is small than the corresponding standard deviation, which better reflects the improvement of the newer spectroscopies because it is less dependent on outliers.

   In more detail, we add the following text at p3,l27:

   The blue and pink symbols indicates collocated data pairs. These are used for further data analysis in this study, whereas all grey data point are discarded. Moreover, to evaluate the quality of the spectral fit for each retrieval, we consider the root-mean-square difference $\sqrt{\frac{1}{L}sum_l(y_{\mathrm{meas},l} - y_{\mathrm{sim},l})^2}$, where index $l$ indicates the $L$ spectral components of the measurement $y_{\mathrm{meas}}$ and its simulation $y_{\mathrm{sim}}$ after convergence of the retrieval. Finally, for a collocated data pair, we determine the corresponding averaged root-mean-square difference.

   For further analysis, we define a set of diagnostic quantities. For each station of our data set, we define the median bias $b_j$ as the median of the difference $\mathrm{XCO}_{ij}^{\mathrm{TROPOMI}} - \mathrm{XCO}_{ij}^{\mathrm{TCCON}}$ between TROPOMI and TCCON XCO daily mean measurements, where index $j$ identifies the station, and $i$ indicates the pair of collocated daily mean values. Also the corespondent median route mean square difference $rms_j$ is determined. To characterize the scatter in the difference between TROPOMI and TCCON data, we consider the percentile difference

   $$\delta P_j = \left|\frac{P_j(84.1) - P_j(15.9)}{2}\right| \tag{1}$$

   of the bias distribution, which corresponds to the standard deviation of normal distributed parameters but it is more robust against outliers. Moreover, the global mean bias $\bar{b}$ is the mean bias of all station biases,

   $$\bar{b} = \frac{1}{n}\sum_{j=1} b_j \tag{2}$$

   with $n$ the number of stations and the station-to-station bias variation is defined as the standard derivation

   $$\bar{\sigma} = \sqrt{\frac{1}{n}\sum_{j=1}^{n}(b_i - \bar{b})}\,. \tag{3}$$

   Accordingly, we updated Fig.3,4, and 5, Table 1 and all numbers in the text.

**Specific comments**

1. *Concerning the collocation criteria with TCCON: In this study data are collocated within a 50km radius and within the same day. Given that average wind speeds in the free troposphere can quickly reach values of 20 to 30 km/hour, a 2 hour collocation window would be a better match for the chosen spatial collocation radius.*

   **not adjusted**

   The reviewer is right that in the case that the variability of the data within the collocation radius is determined by dynamics, representation errors can be reduced when harmonize collocation radius and temporal collocation window. However, when the variability in the data is due to other causes, this argument may not hold. We reanalyzed our data with a collocation time of 2 hours between TCCON and TROPOMI and found no significant differences with respect to the reported analysis. For comparability with our first study on TROPOMI CO data, we therefore decided to use the daily means in our study.

2. *Fig 5: The caption mentions that it is like Fig 3. Yet the rms plot is replaced by a number of coincidences plot. This should be mentioned.*

   **adjusted**

   We added the following sentence to the caption of Figure 5:

[revised manuscript text omitted]

We would like to thank the reviewer for the constructive comments that aided us to improve our manuscript. In this document we provide our replies to the reviewer's comments. The original comments made by the reviewer are numbered and typeset in italic and bold face font. Following every comment we give our reply. Here line numbers, page numbers and figure numbers refer to the original version of the manuscript, if not stated differently. Additionally, the revised version of the manuscript is added.

**Major comments**

1. *The authors should be careful to exactly define what is meant by certain terms when they are introduced in the text. For instance what is implied by the average bias and standard deviation of the bias? Are all collocated data pairs averaged regardless of their station origin (implying a stronger impact on the average bias for TCCON stations that have many collocations (for instance East Trout Lake) versus stations that have few (Edwards)) or is it the average of the individual station biases? From the caption in Fig 3 I assume the later but this should be mentioned in the main text.*

2. *Similarly for the station-to-station variability of the bias (again I assume std of the bias over different stations based on the figures) To evaluate the significance of a bias, the standard error or confidence interval is a far more useful metric than the standard deviation of the bias. Particularly when evaluating different products.*

**adjusted**

We agree with both comments of the reviewer. Therefore, we adapted our study such that our statistical analysis is based on the median bias and the percentile difference $\delta P$ (see definition below) to quantify the scatter in the biases per station. The overall conclusions of our study remains the same. Here, the data scatter $\delta P$ is small than the corresponding standard deviation, which better reflects the improvement of the newer spectroscopies because it is less dependent on outliers.

In more detail, we add the following text at p3,l27:

The blue and pink symbols indicates collocated data pairs. These are used for further data analysis in this study, whereas all grey data point are discarded. Moreover, to evaluate the quality of the spectral fit for each retrieval, we consider the root-mean-square difference $\sqrt{\frac{1}{L}sum_l(y_{meas,l} - y_{sim,l})^2}$, where index $l$ indicates the $L$ spectral components of the measurement $y_{meas}$ and its simulation $y_{sim}$ after convergence of the retrieval. Finally, for a colocated data pair, we determine the corresponding averaged root-mean-square difference.

For further analysis, we define a set of diagnostic quantities. For each station of our data set, we define the median bias $b_j$ as the median of the difference $XCO_{i,j}^{TROPOMI} - XCO_{i,j}^{TCCON}$ between TROPOMI and TCCON XCO daily mean measurements, where index $j$ identifies the station, and $i$ indicates the pair of collocated daily mean values. Also the correspondent median route mean square difference $rms_j$ is determined. To characterize the scatter in the difference between TROPOMI and TCCON data, we consider the percentile difference

$$\delta P_j = \left| \frac{P_j(84.1) - P_j(15.9)}{2} \right| \tag{1}$$

of the bias distribution, which corresponds to the standard deviation of normal distributed parameters but it is more robust against outliers. Moreover, the global mean bias $\bar{b}$ is the mean bias of all station biases,

$$\bar{b} = \frac{1}{n}\sum_{j=1}^{n} b_j \tag{2}$$

with n the number of stations and the station-to-station bias variation is defined as the standard derivation

$$\bar{\sigma} = \sqrt{\frac{1}{n}\sum_{j=1}^{n}(b_i - \bar{b}_i)} . \tag{3}$$

Accordingly, we updated Fig.3,4, and 5, Table 1 and all numbers in the text.

**Specific comments**

1. *Concerning the collocation criteria with TCCON: In this study data are collocated within a 50km radius and within the same day. Given that average wind speeds in the free troposphere can quickly reach values of 20 to 30 km/hour, a 2 hour collocation window would be a better match for the chosen spatial collocation radius.*

**not adjusted**

The reviewer is right that in the case that the variability of the data within the collocation radius is determined by dynamics, representation errors can be reduced when harmonize collocation radius and temporal collocation window. However, when the variability in the data is due to other causes, this argument may not hold. We reanalyzed our data with a collocation time of 2 hours between TCCON and TROPOMI and found no significant differences with respect to the reported analysis. For comparability with our first study on TROPOMI CO data, we therefore decided to use the daily means in our study.

2. *Fig 5: The caption mentions that it is like Fig 3. Yet the rms plot is replaced by a number of coincidences plot. This should be mentioned.*

**adjusted**

We added the following sentence to the caption of Figure 5:

[revised manuscript text omitted]

---

## Author Response (AR2)

**author comments on the manuscript "Improving the TROPOMI CO data product: update of the spectroscopic database and destriping of single orbits", Editor**

We want to thank the Editor Hartmut Boesch, for the quick decision. In this document we provide our replies to his comments. The original comments made by him are numbered and typeset in italic and bold face font. Following every comment we give our reply. Here line numbers, page numbers and figure numbers refer to the previous version of the manuscript, if not stated differently. Additionally, the revised version of the manuscript is added.

1. *it might be good to add a quick statement on the choice of 15.9 and 84.1 percentiles (which I assume is used as it corresponds to +/1 1-sigma around the mean for a normal curve) in l.16 on page 4*

   **adjusted**

   we changed the sentence at p4, l16 from:
   "which corresponds to the standard deviation of normal distributed parameters but it is more robust against outliers." to
   "which corresponds to the standard deviation of normal distributed parameters but it is more robust against outliers. Hence, the choice of 84.1 and 15.9 percentiles would be the ±1 1-sigma around the mean for a normal curve. "

2. *can you please replace 'sum' on l8, page 4 with a proper sum symbol.*

   **adjusted**

3. *- in your response to the first comment of reviewer 3, you say: Here, the bias between TROPOMI CO and the TCCON measurements was used to estimate the product accuracy and the scatter in the difference between both measurements indicated an upper boundary for the accuracy and precision of the TROPOMI instrument. -¿In relation to the observed scatter in the difference, do you really mean 'accuracy and precision' or only 'precision' ?*

   **adjusted**

   Indeed, that was wrongly stated. We changed the sentence p2,l23 from:
   " Here, the bias between TROPOMI and the TCCON CO measurements was used to estimate the product accuracy and the scatter in the difference between both measurements indicated an upper boundary for the accuracy and precision of the TROPOMI instrument. " to
   " Here, the bias between TROPOMI and the TCCON CO measurements was used to estimate the product accuracy and the scatter in the difference between both measurements indicated an upper boundary for the precision of the TROPOMI instrument. "

**Additional changes**

Dietrich Feist requested a change of his affiliations. This is updated in the new version of the manuscript.

[revised manuscript text omitted]